# Detection of Atmospheric Rivers with Inline Uncertainty Quantification: TECA-BARD v1.0.1

Travis A. O'Brien[1,2], Mark D. Risser[2], Burlen Loring[3], Abdelrahman A. Elbashandy[3], Harinarayan Krishnan[3], Jeffrey Johnson[4], Christina M. Patricola[2,5], John P. O'Brien[2,6], Ankur Mahesh[2], Prabhat[7], Sarahí Arriaga Ramirez[2,8], Alan M. Rhoades[2], Alexander Charn[2,9], Héctor Inda Díaz[2,8], and William D. Collins[2,9]

[1]Dept. of Earth and Atmospheric Sciences, Indiana University, Bloomington, Indiana, USA
[2]Climate and Ecosystem Sciences Division, Lawrence Berkeley National Lab, Berkeley, California, USA
[3]Computational Sciences Division, Lawrence Berkeley National Lab, Berkeley, California, USA
[4]Cohere Consulting LLC, Seattle, WA, USA
[5]Dept. of Geological and Atmospheric Sciences, Iowa State University, Ames, Iowa, USA
[6]Dept. of Earth and Planetary Science, University of California, Santa Cruz, California, USA
[7]National Energy Research Scientific Computing Center, Lawrence Berkeley National Lab, Berkeley, California, USA
[8]Dept. of Land, Air and Water Resources, University of California, Davis, California, USA
[9]Dept. of Earth and Planetary Science, University of California, Berkeley, California, USA

**Correspondence:** Travis A. O'Brien (obrienta@iu.edu)

**Abstract.** It has become increasingly common for researchers to utilize methods that identify weather features in climate models. There is an increasing recognition that the uncertainty associated with choice of detection method may affect our scientific understanding. For example, results from the Atmospheric River Tracking Method Intercomparison Project (ARTMIP) indicate that there are a broad range of plausible atmospheric river (AR) detectors, and that scientific results can depend on the algorithm used. There are similar examples from the literature on extratropical cyclones and tropical cyclones. It is therefore imperative to develop detection techniques that explicitly quantify the uncertainty associated with the detection of events. We seek to answer the question: given a 'plausible' AR detector, how does uncertainty in the detector quantitatively impact scientific results? We develop a large dataset of global AR counts, manually identified by a set of 8 researchers with expertise in atmospheric science, which we use to constrain parameters in a novel AR detection method. We use a Bayesian framework to sample from the set of AR detector parameters that yield AR counts similar to the expert database of AR counts; this yields a set of 'plausible' AR detectors from which we can assess quantitative uncertainty. This probabilistic AR detector has been implemented in the Toolkit for Extreme Climate Analysis (TECA), which allows for efficient processing of petabyte-scale datasets. We apply the TECA Bayesian AR Detector, TECA-BARD v1.0.1, to the MERRA2 reanalysis and show that the sign of the correlation between global AR count and El Niño Southern Oscillation depends on the set of parameters used.

## 1 Introduction

There is a growing body of literature in which researchers decompose precipitation and other meteorological processes into constituent weather phenomena, such as tropical cyclones, extratropical cyclones, fronts, mesoscale convective systems, and

atmospheric rivers (e.g., Kunkel et al., 2012; Neu et al., 2013; Walsh et al., 2015; Schemm et al., 2018; Zarzycki et al., 2017; Wehner et al., 2018). Research focused on atmospheric rivers (ARs) in particular has contributed a great deal to our understanding of the water cycle (Zhu and Newell, 1998; Sellars et al., 2017), atmospheric dynamics (Hu et al., 2017), precipitation variability (Dong et al., 2018), precipitation extremes (Leung and Qian, 2009; Dong et al., 2018), impacts (Neiman et al., 2008; Ralph et al., 2013, 2019a), meteorological controls on the cryosphere (Gorodetskaya et al., 2014; Huning et al., 2017, 2019), and uncertainty in projections of precipitation in future climate change scenarios (Gershunov et al., 2019b).

Over the past decade, there has been a growth in the number of methods used to detect ARs, and in the last five years there has been a growing recognition that uncertainty in AR detection may impact our scientific understanding; the Atmospheric River Tracking Method Intercomparison Project (ARTMIP) was created to assess this impact (Shields et al., 2018). Through a series of controlled, collaborative experiments, results from ARTMIP have shown that at least some aspects of our understanding of AR-related science indeed depend on detector design (Shields et al., 2018; Rutz et al., 2019). Efforts related to ARTMIP have similarly shown that some aspects of AR-related science depend on the detection algorithm used (Huning et al., 2017; Ralph et al., 2019b).

ARTMIP has put significant effort into quantifying uncertainty, and the community is poised to imminently produce several important papers on this topic. It would be impractical to perform ARTMIP-like experiments for every AR-related science question that arises, which raises the question of how best to practically deal with uncertainty in AR detection.

This uncertainty arises because there is no theoretical and quantitative definition of an AR. Only recently did the community come to a consensus on a qualitative definition (Ralph et al., 2018). In order to do quantitative science related to ARs, researchers have had to independently form quantitative methods to define ARs (Shields et al., 2018). Existing AR detection algorithms in the literature are predominantly heuristic: e.g., they consist of a set of rules used to isolate ARs in meteorological fields. Inevitably, heuristic algorithms also contain unconstrained parameters (e.g., thresholds). Across the phenomenon-detection literature (ARs and other phenomena), the prevailing practice is for researchers to use expert judgement to select these parameters. The two exceptions of which the authors are aware is that of Zarzycki and Ullrich (2017) and of Vishnu et al. (2020), who use an optimization method to determine parameters for a tropical cyclone (TC) detector and monsoon depression detector respectively.

Even if one were to adopt a similar optimization framework for an AR detector, this still would not address the issue that uncertainty in AR detection can qualitatively affect scientific results. This sort of problem has motivated the use of formal uncertainty quantification frameworks, in which an ensemble of 'plausible' AR detectors are run simultaneously. However, these frameworks need data against which to assess the plausibility of a given AR detector. Zarzycki and Ullrich (2017) and Vishnu et al. (2020) were able to take advantage of existing, human-curated track datasets. No such dataset exists for ARs.

A key challenge for developing such a dataset is the human effort required to develop it. The best type of dataset would presumably be one in which experts outline the spatial footprints of ARs, such as the ClimateNet dataset described in the forthcoming manuscript by Prabhat et al. (2020). At the time that the work on this manuscript started, the ClimateNet dataset did not yet exist, and we considered that the simpler alternative would be to identify the number of ARs in a set of given

meteorological fields. Even though a dataset of AR counts is perhaps less informative than a dataset of AR footprints, we hypothesize that such a dataset could serve to constrain the parameters in a given AR detector.

This manuscript addresses the dual challenges of uncertainty quantification and optimization: we develop a formal Bayesian framework for sampling 'plausible' sets of parameters from an AR detector, and we develop a database of AR counts with which to constrain the Bayesian method. We provide a general outline for the Bayesian framework as well as a specific implementation: the Toolkit for Extreme Climate Analysis Bayesian AR Detector version 1.0.1 (TECA-BARD v1.0.1; Section 2). We show that TECA-BARD v1.0.1 performs comparably to an ensemble of algorithms from ARTMIP and that it emulates the counting statistics of the contributors who provided AR counts (Section 3). We demonstrate that answers to the question *Are there more ARs during El Niño events?* depends qualitatively on the set of detection parameters (Section 4).

## 2   The Bayesian approach

### 2.1   Overview

We start with a general description of how a Bayesian framework, in combination with a dataset of AR counts, can be applied to an AR detector. We consider a generic heuristic detection algorithm with tunable parameters $\boldsymbol{\theta}$ (thresholds, etc.) that, when given an input field $\boldsymbol{Q}$ (e.g., integrated vapor transport, IVT), can produce a count of ARs within that field. For compactness, we will represent this heuristic algorithm and subsequent counting as a function $f(\boldsymbol{\theta}|\boldsymbol{Q})$. That is, for a given field $\boldsymbol{Q}$ and a specific choice of tuning parameters, $f$ returns the number of detected ARs in $\boldsymbol{Q}$.

Further, we assume that we have a dataset of $M$ actual AR counts, denoted by $\boldsymbol{N}$, associated with a set of independent input fields (i.e., generated by an expert counting the ARs; see Section 2.2): $\{(N_i, \boldsymbol{Q}_i) : i = 1, \ldots, M\}$. With a quantitatively defined prior on the tunable parameters $p_\theta(\boldsymbol{\theta})$, we can use Bayes' theorem to define the posterior probability of $\boldsymbol{\theta}$ given the AR counts $\boldsymbol{N}$ and input fields $\boldsymbol{Q}$:

$$p(\boldsymbol{\theta} \mid \boldsymbol{N}, \boldsymbol{Q}) \propto \frac{\left( \prod_{i=1}^{M} \mathcal{L}(N_i \mid \boldsymbol{\theta}, \boldsymbol{Q}_i) \right) \cdot p_\theta(\boldsymbol{\theta})}{p_N(\boldsymbol{N})} \tag{1}$$

We propose to base the likelihood $\mathcal{L}$ on counts from the heuristic model $N_i' = f(\boldsymbol{\theta}, \boldsymbol{Q}_i)$. We model $\mathcal{L}$ as a normal distribution centered on $N_i'$:

$$\mathcal{L}(N_i \mid \boldsymbol{\theta}, \boldsymbol{Q}_i) = \mathcal{N}(N_i \mid N_i', \sigma), \tag{2}$$

where $\sigma$ is a nuisance parameter that is ultimately integrated over. While the normal distribution is typically assigned to a continuous (real-valued) variable, here we simply use it as a quantitative way to minimize the squared error between each $N_i$ and $N_i'$.

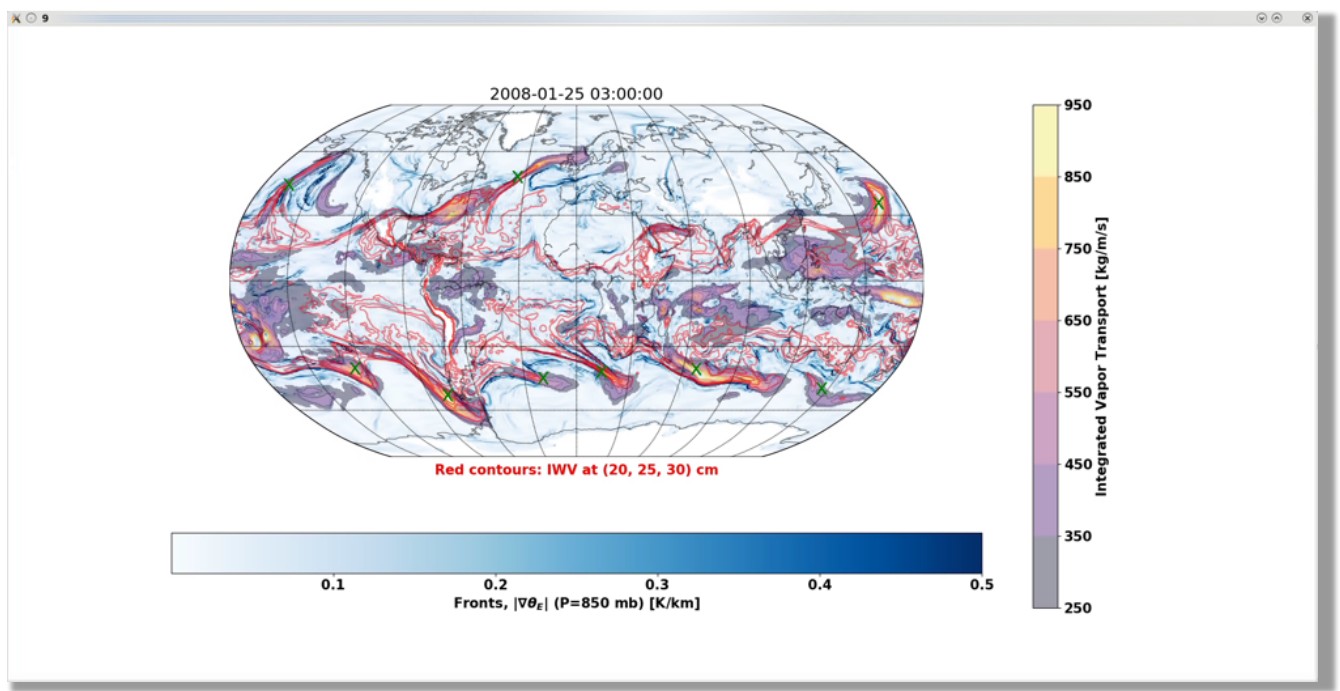

**Figure 1.** An example screenshot of a 3-hourly time slice of MERRA2 derived integrated vapor transport using a graphical user interface (GUI) that eight co-authors of this manuscript used to count ARs for a training dataset. The expert is presented with an overlay of information about IVT (purple-yellow shading), integrated water vapor (red contours), and the magnitude of the 850 hPa equivalent potential temperature gradient (blue shading).

## 2.2 A Database of Expert AR Counts

In order to constrain a Bayesian AR detection algorithm, we developed a database of global AR counts. We designed a simple graphical user interface (GUI) that displays a meteorological plot, as shown in Figure 1, for a given instance of time. The meteorological plot overlays information about IVT, integrated water vapor, and the magnitude of gradients in 850 hPa equiv-

5  alent potential temperature (indicative of fronts); the sample image in Figure 1 shows a screenshot of this information as it is presented to the expert contributors. Times are chosen randomly within the years 2008 and 2009, which were chosen to correspond to the time period associated with the Year of Tropical Convection (Waliser et al., 2012). The interface allows a user to enumerate ARs within a given field by clicking the mouse in the vicinity of an AR. A graphical indicator (a small, green 'X') is left in the location of the mouse click, which allows the user to visually assess whether they have adequately accounted

10  for all ARs in a given field before proceeding to the next image. The GUI-relative coordinates of each click is recorded in the metadata, which allows approximate reconstruction of the geophysical location of each indicated AR. The location information is not used in constraining the Bayesian AR detection algorithm, though we do use it for understanding differences among expert contributors.

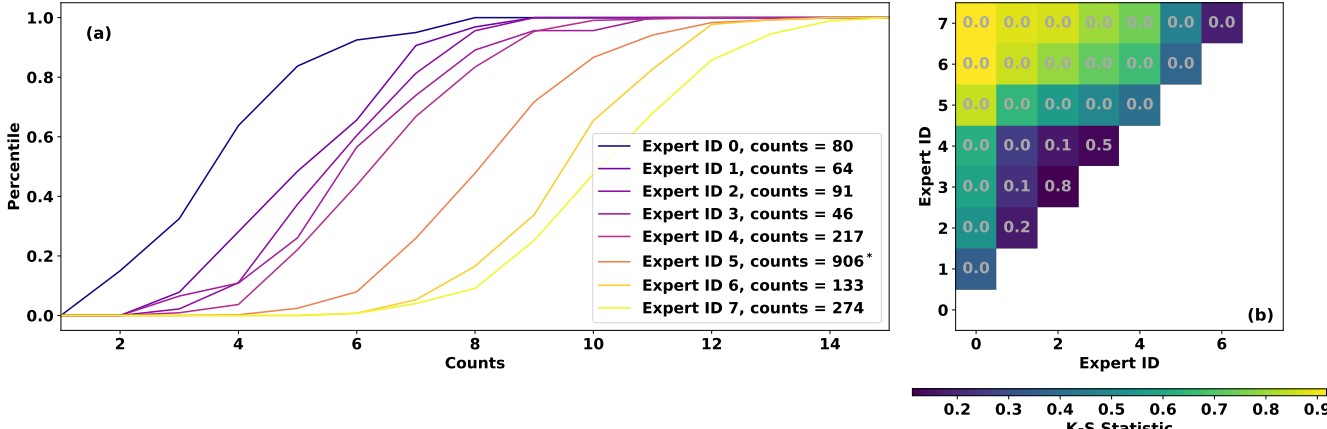

**Figure 2.** (a) Cumulative distributions of expert counts. (b) Two-sample Kolmogorov-Smirnov test statistics among Expert IDs. Gray text indicates the p-value; low values indicate that sets of expert counts likely have different distributions. *Expert ID 5 provided 906 sets of counts, but only 250 are used in the MCMC sampling stage (see Section 2.4.1) due to computational considerations.

Eight of the co-authors of this manuscript (see Acknowledgements) contributed counts via this GUI, and the counts differ substantially. Each contributor counted ARs in at least 30 random time slices, with contributions ranging between 46 and 906 time slices (see Figure 2a). Figure 2a shows that the number of ARs counted varies by nearly a factor of 3 among contributors: the most 'restrictive' expert identifies a median of 4 ARs, while the most 'permissive' expert identifies a median of 11 ARs.

Contributors are assigned an identification number according to the mean number of ARs counted, with the lowest 'Expert ID' (zero) having the lowest mean count and Expert ID 7 having the highest. Differences among the cumulative distributions shown in Figure 2 are mostly statistically significant, according to a suite of pair-wise Kolmogorov-Smirnov tests (Figure 2b). Counts from Expert IDs 1, 2, and 3 are mutually statistically indistinguishable at the 90% confidence level. Expert IDs 3 and 4 are likewise statistically indistinguishable, though 4 differs significantly from 1 and 2.

The differences among expert contributors leads one to wonder whether they are counting the same meteorological phenomenon, and cross-examination suggests that they are. There are a number of instances where, by chance, three experts counted ARs in the same time slice. Intercomparison of the approximate AR locations in these multiply-counted time slices (not shown) indicates that the most restrictive contributors tend to identify the same meteorological features as the most permissive contributors. The ARs identified by restrictive contributors are a subset of those identified by the permissive contributors.

These differences present two methodological challenges: (1) differences among the expert contributors will likely lead to different groups of parameter sets in a Bayesian algorithm; and (2) there is nearly an order-of-magnitude spread among the number of time slices contributed by each expert, which would lead to over-representation of the contributors with the highest number of time slices (e.g., Expert ID 5 contributed 906 counts; Figure 2a). We opt to treat all expert contributions as equally plausible, given that there is no *a priori* constraint (e.g., physical constraint or otherwise) on the number of ARs globally. Both

challenges can be addressed simply by doing the Bayesian model fitting separately for each expert and then pooling parameters in the final stage; this procedure is described in more detail in Section 2.4.1.

## 2.3 A Specific Implementation - TECA-BARD v1.0.1

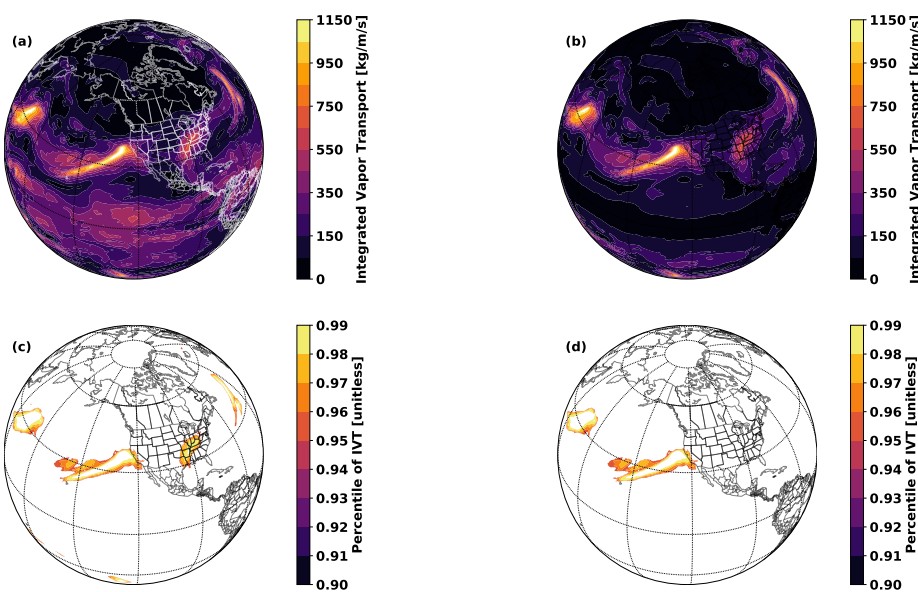

**Figure 3.** Illustration of the steps in TECA AR v1.0.1 with $\Delta y = 15°$N, $P = 0.95$, and $A_{\min} = 1 \cdot 10^{12}$ m$^2$: (a) the input field, integrated vapor transport (IVT); (b) IVT after application of a $\Delta y = 15°$N tropical filter (IVT'); (c) IVT' (converted to percentile) after application of the percentile filter; and (d) after application of the minimum area filter.

We propose here a specific implementation of an AR detector on which to test the Bayesian method. For the sake of par-
5  simony, this initial detector includes only three main criteria: contiguity above a threshold, size, and location. The detector utilizes a spatially filtered version of the IVT field, IVT' (defined toward the end of this paragraph), and in this specific implementation it seeks contiguous regions within each 2D field that are above a time-dependent threshold, where the threshold is defined as the $P^{\text{th}}$ percentile of that specific IVT' field. This follows the motivation of Shields and Kiehl (2016), who utilize a time-dependent threshold in order to avoid ARs becoming arbitrarily larger as water vapor mixing ratios increase in the
10  atmosphere due to global warming. The contiguous regions must have an area that is greater than a specified threshold $A_{\min}$. In order to avoid large contiguous regions in the tropics, associated with the intertropical convergence zone (ITCZ), the IVT field is spatially filtered as

$$\text{IVT'}(y,x) = \left(1 - e^{-2\ln 2 \cdot \frac{y^2}{\Delta y^2}}\right) \cdot \text{IVT}(y,x), \tag{3}$$

where $(y, x)$ are spatial coordinates (latitude and longitude respectively), and $\Delta y$ is the half-width at half-maximum of the filter. The filter essentially tapers the IVT field to 0 in the tropics, within a band of approximate width $\Delta y$. Table 1 summarizes the free parameters in this AR detector, and Figure 3 illustrates the stages of the detection algorithm

**Table 1.** Parameters, ranges, and priors in the AR detector

| Parameter | Description | Range |
|---|---|---|
| $P$ | Percentile threshold for IVT' | $(0.8, 0.99)$ |
| $A_{\min}$ | Minimum area of contiguous region | $(1 \cdot 10^{11}, 5 \cdot 10^{12})\,\mathrm{m}^2$ |
| $\Delta y$ | Zonal half-width-at-half-maximum of tropical filter | $(5, 25)\,^\circ\mathrm{N}$ |

Table 1 also presents the prior ranges that we deem plausible for the parameter values; justification of these ranges follows. For $\Delta y$, the filter should efficiently damp the ITCZ toward 0. Though the ITCZ is relatively narrow, it migrates significantly throughout the annual cycle, so we use a minimum threshold of $5°$ as the lower bound. The filter should not extend so far north that it damps the midlatitudes, which is where the ARs of interest are located; hence we use an upper bound of $25°$, which terminates the filter upon entering the midlatitudes. For $A_{\min}$, we use an order-of-magnitude range based on experience in viewing ARs in meteorological data; for reference, we note that ARs are often of a size comparable to the state of California. $1 \cdot 10^{11}\,\mathrm{m}^2$ is approximately one quarter of the area of the state of California, which is likely on the too-small side, and $5 \cdot 10^{12}$ is approximately 6 times the area of California. For $P$, we note that the threshold is linked to the fraction of the planetary area that ARs cover in total. We use 20% of the planetary area as an upper bound ($P = 0.8$), and 0.1% as a lower bound ($P = 0.99$). The actual area covered by ARs of course depends both on the typical area of ARs and the typical number. If we assume that there are $\mathcal{O}(10)$ ARs occurring globally at any time, and they have a size $\mathcal{O}(10^{12}\,\mathrm{m}^2)$, then they would cover $\mathcal{O}(10\%)$ of the planetary area ($P = 0.9$) as postulated by one of the earliest AR manuscripts (Zhu and Newell, 1998).

We refer to this specific implementation of AR detector, in terms of the AR counts that it yields, as $F_3$, such that $N_i' = F_3(P, A_{\min}, \Delta y | \boldsymbol{Q_i})$. We use a half-Cauchy prior for $\sigma$ (Equation 2), following Gelman (2006): $P_\sigma = (2/\pi s)\left(1 + (\sigma/s)^2\right)^{-1}$, and we fix the scale parameter $s$ at a large value of 10, which permits a wide range of $\sigma$ values. $\sigma$ is the parameter controlling the width of the likelihood function, which effectively controls how far the detected counts $N_i'$ can deviate from the expert counts $N_i$ before the likelihood function indicates that a given choice of $(P, A_{\min}, \Delta y)$ is unlikely compatible with the expert data; we treat it as a nuisance parameter in our model. Given this and a choice of a uniform prior for all three parameters, and an assumption that the prior distributions are independent, this leads to a concrete Bayesian model for the posterior distribution of the parameter set $(P, A_{\min}, \Delta y)$:

$$p(P, A, \Delta y \mid \boldsymbol{N}, \boldsymbol{Q}) \propto \left( \prod_{i=1}^{M} \mathcal{N}\left(N_i \mid F_3(P, A_{\min}, \Delta y | \boldsymbol{Q_i}), \sigma\right) \right) \cdot \frac{2}{\pi s} \frac{1}{[1 + (\sigma/s)^2]} \tag{4}$$

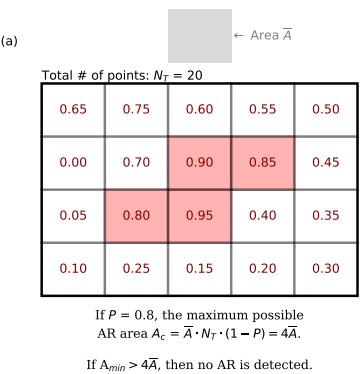

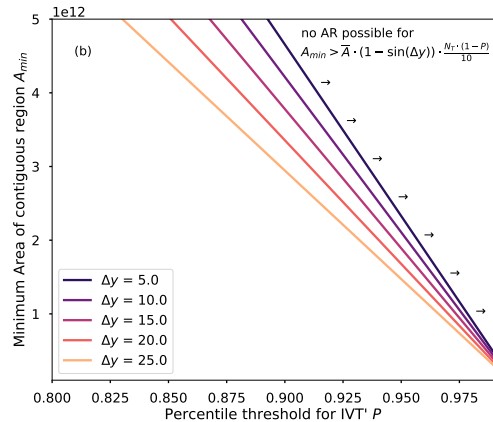

**Figure 4.** Illustration of the geometric constraints applied to the prior distribution of parameters $P$, $A_{\min}$, and $\Delta y$. (a) A diagram depicting the interaction between percentile threshold $P$ and minimum area $A_{\min}$. Red text depicts hypothetical IVT' percentile values for individual gridboxes (gray boxes); boxes with $P$ above 0.8 are shaded in red. (b) Visualization of Equation 5 for select values of $\Delta y$, and annotation indicating regions of the $A_{\min} - P - \Delta y$ parameter space that are *a priori* implausible because they would yield no AR detections.

### 2.3.1 Geometrically Constraining the Prior

The prior parameter ranges in Table 1 provide plausible prior ranges for the detector parameters, but there are some areas within this cube of parameters that we can *a priori* assert are highly improbable due to geometric considerations. This is necessary in order to avoid the Markov Chain Monte Carlo algorithm (see Section 2.4) from having points that initialize and get 'stuck' in

5     regions of the parameter space that do not yield ARs.

By definition, the percentile threshold $P$ will select $N_c = (1 - P) \cdot N_T$ points out of the total $N_T$ points in the input field. If we approximate the area of all individual grid cells (ignoring for simplicity the latitudinal dependence) as $\overline{A}$, then the total area of cells above the percentile threshold will be $A_c = \overline{A} N_T (1 - P)$. By deduction, in order for any AR to be detected, the total area of grid cells above the threshold $P$ must be as large or larger than the minimum-area threshold $A_{\min}$ for contiguous blobs

10     above the percentile threshold: i.e., if $A_{\min} > A_c$, then no AR detections are possible. We assert that parameter combinations that prohibit AR detections are implausible, and therefore the prior should be equal to 0 in such regions of parameter space. This condition effectively defines a line in the $A_{\min}$ vs $P$ plane, where the prior is 0 to the right of the curve:

$$A_{\min} = \overline{A} N_T (1 - P)$$

Figure 4a depicts the geometric relationship between $A_{\min}$ and $P$: as $P$ increases, the maximum permissible value of $A_{\min}$ decreases.

This idea can be expanded further by noting that the latitude filter effectively sets values near a band $2\Delta y$ close to 0. If we assume that all points within $2\Delta y$ of the equator are effectively removed from consideration, then the total number of points under consideration $N_T$ should be reduced by the fraction $f$ of points that are taken out by the filter. In the latitudinal direction, cell areas are only a function of latitude $y$ ($\cos(y)$ specifically), so with the above assumption, $f$ can be approximated simply as:

$$f = 1 - \frac{\int_{-\Delta y}^{\Delta y} \cos(y)\, dy}{\int_{-\pi/2}^{\pi/2} \cos(y)\, dy} = 1 - \sin(\Delta y).$$

With this, the number of cells passing the threshold test shown in Figure 3c will be approximately $N_c' = f \cdot N_T \cdot (1 - P)$. If we assume that there are $\mathcal{O}(10)$ ARs at any given time, then there are typically at most $N_c'/10$ grid cells per AR. We tighten the constraint to assert that these conditions should lead to ARs that typically have more than 1 grid cell per AR. The assertion that ARs should typically consist of more than 1 grid cell is only valid if $\overline{A}$ is substantially less than the area of a typical AR. We use MERRA2 reanalysis, with $\overline{A} = 2.5 \cdot 10^9 m^2$, which is almost two orders of magnitude smaller than the lower bound on the minimum AR size of $1 \cdot 10^{11} m^2$ (Table 1), so even the smallest possible ARs detected will consist of $\mathcal{O}(100)$ grid cells. This assertion might need to be revisited if one were to train the Bayesian model on much lower resolution data. This leads to a formulation of the prior constraint that depends on the value of the latitude filter, such that only parameter combinations that satisfy the following inequality are permitted:

$$A_{\min} \leq \overline{A} \cdot (1 - \sin(\Delta y)) \cdot \frac{N_T \cdot (1 - P)}{10}. \tag{5}$$

We modify the uniform prior to be equal to 0 outside the surface defined in Equation 5 (to the right of the $A_{\min}(P, \Delta y)$ lines shown in Figure 4b).

## 2.4 Markov Chain Monte Carlo Sampling

We use an affine-transformation-invariant Markov Chain Monte Carlo (MCMC) sampling method (Goodman and Weare, 2010), implemented in Python by Foreman-Mackey et al. (2013) (emcee v2.2.1[1]), to approximately sample from the posterior distribution described in Equation 4. We utilize 128 MCMC 'walkers' (semi-independent MCMC chains) with starting positions sampled uniformly from the parameter ranges shown in Table 1. Parameter values outside the parameter-surface described by Equation 5 are rejected and randomly sampled until all initial parameter sets satisfy Equation 5.

The MCMC algorithm essentially finds sets of parameters for which TECA-BARD yields sets of AR counts that are close (in a least-squares sense) to the input set of expert counts described in Section 2.2. Within an MCMC step, each walker proposes a new set of parameters. Each MCMC walker runs the TECA-BARD algorithm described in Section 2.3, for its set of proposed

---

[1]https://github.com/dfm/emcee/releases/tag/v2.2.1

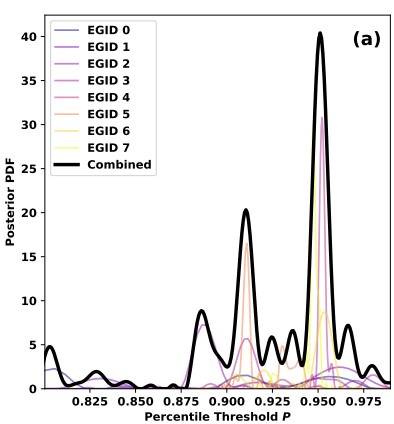 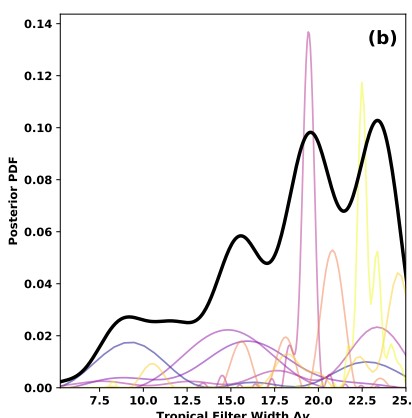 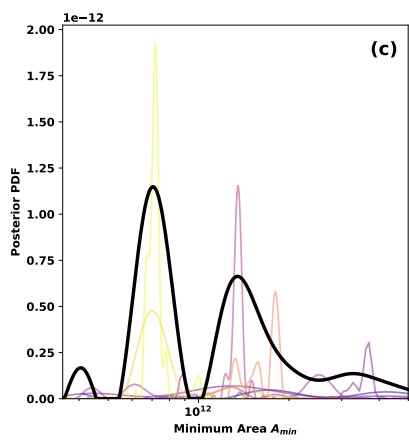

**Figure 5.** Posterior marginal distributions of parameters for each Expert Group ID (EGID) $p_j(\boldsymbol{\theta})$ (colored curves) and for the full, combined posterior $p(\boldsymbol{\theta})$ (black curve): (a) percentile threshold for IVT' $P$, (b) zonal half-width-at-half-maximum of the tropical filter $\Delta y$, and (c) minimum area of contiguous regions $A_{\min}$. Posterior distributions for EGIDs are scaled by $\frac{1}{8}$, consistent with Equation 6

parameter values, on the IVT field ($Q_i$) from all time slices in MERRA2 for which there are expert counts $N_i$; TECA-BARD ($F_3$ in Equation 4) returns the global number of ARs $N_i'$ for each time slice. The sets of expert counts and TECA-BARD counts are provided as input to Equation 2, which is then used in Equation 1 to evaluate the posterior probability of the proposed parameters. The proposed parameter is then either accepted or rejected following the algorithm outlined by Foreman-
5   Mackey et al. (2013). Parameters with higher posterior probabilities generally have a higher chance of being accepted. The accept/reject step has an adjustable parameter ($a$ in Equation 10 of Foreman-Mackey et al., 2013), which we set to a value of 2, following Goodman and Weare (2010). Sensitivity tests with this value showed little qualitative change in the output of the MCMC samples.

We run all 128 MCMC walkers for 1,000 steps and extract MCMC samples from the last step. We use an informal process to
10   assess equilibration of the MCMC sampling chains: we manually examine traces (the evolution of parameters within individual walker chains). The traces reach a dynamic steady-state after $\mathcal{O}(100)$ steps, so we expect that the chains should all be well-equilibrated by 1,000 steps. We ran a brute-force calculation of the posterior distribution on a regularly-spaced grid of parameter values (not shown) to verify that the MCMC algorithm is indeed sampling correctly from the posterior distribution, which further evinces that the MCMC process has reached equilibrium by the 1,000[th] step.

### 2.4.1 Expert Groups and Multimodality

In order to address the challenges posed by having AR count datasets that differ significantly among expert contributors (described in Section 2.2), we develop a separate posterior model for each Expert ID $j$: $p_j(\boldsymbol{\theta} \mid \boldsymbol{N}_j, \boldsymbol{Q})$. The final model is a normalized, unweighted sum of posterior distributions from each Expert ID:

$$p(\boldsymbol{\theta} \mid \boldsymbol{N}, \boldsymbol{Q}) = \frac{1}{8} \cdot [p_0(\boldsymbol{\theta} \mid \boldsymbol{N}_0, \boldsymbol{Q}) + p_1(\boldsymbol{\theta} \mid \boldsymbol{N}_1, \boldsymbol{Q}) + \ldots + p_7(\boldsymbol{\theta} \mid \boldsymbol{N}_7, \boldsymbol{Q})] \tag{6}$$

Practically speaking, we achieve this by running the MCMC integration separately for each Expert ID and then combining the MCMC samples together. TECA BARD v1.0.1 uses each of the 128 MCMC samples generated for each Expert ID; with 8 Expert IDs, this gives a total of 1,024 sets of parameters used in TECA BARD v1.0.1. The samples are stored in an input parameter table such that parameters from the same Expert ID are contiguous, which allows post hoc grouping of results by

Expert ID. We refer to these groups by their *Expert Group IDs*, which correspond to data from each Expert ID used in the MCMC integration. Figure 5 shows marginal distributions of the TECA-BARD v1.0.1 parameters.

Hereafter, we use two similar and related, but distinct, terms:

  – **Expert ID**: the identification number of a given contributor to the expert count database. EIDs are assigned in order of the mean number of ARs that the expert typically counts in a given timestep.

– **EGID** - Expert *Group* ID: the identification number of groups of posterior parameters obtained by training the Bayesian model on expert counts contributed by the corresponding EID (see Equation 6).

The posterior distributions exhibit multimodality: both in the individual EGID posterior distributions and in the combined posterior distributions shown in Figure 5. This multimodality arises as a consequence of three factors: (1) parameter-dependence of the counts generated by the AR detector, which depends on the underlying IVT field being analyzed, (2) variabil-

ity in the counts from each expert, and (3) the addition of posterior distributions from each EGID–each having their own distinct modes. To illustrate how the first two factors lead to inherent multimodality, Figures 6a–h show the dependence of the counts generated by the AR detector on the percentile and minimum area thresholds (orange contours): $F_3(P, A_{\min} \mid \Delta y = 15, \boldsymbol{Q_i})$ for eight random IVT fields $\boldsymbol{Q_i}$. $F_3$ exhibits similar qualitative dependence on $P$ and $A_{\min}$ for all eight cases: AR count tends to be high for low values of both $P$ and $A_{\min}$, and it tends to be low when both $P$ and $A_{\min}$ are high (see Section 2.3.1 for

an explanation of the geometric relationship that leads to this behavior). Aside from this general qualitative agreement, the fine-scale details of the dependence of $F_3$ on $P$ and $A_{\min}$ depends strongly on the actual IVT field (compare Figures 6 (a) and (f) for example). Non-monotonic dependence of $F_3$ on the input parameters arises, for example, from ARs merging as $P$ is reduced or splitting as $P$ is increased (merging reduces the count, splitting increases the count). It is not surprising that the number of ARs detected depends simultaneously on the parameters controlling the AR detector and the IVT field in which

ARs are being detected.

The number of ARs counted by a given expert also depends on the given IVT field. The bold orange contour in Figures 6a–h shows the number of ARs counted by Expert ID 6; $N_i'$ is a single scalar number for each field $\boldsymbol{Q_i}$, and we show it as a

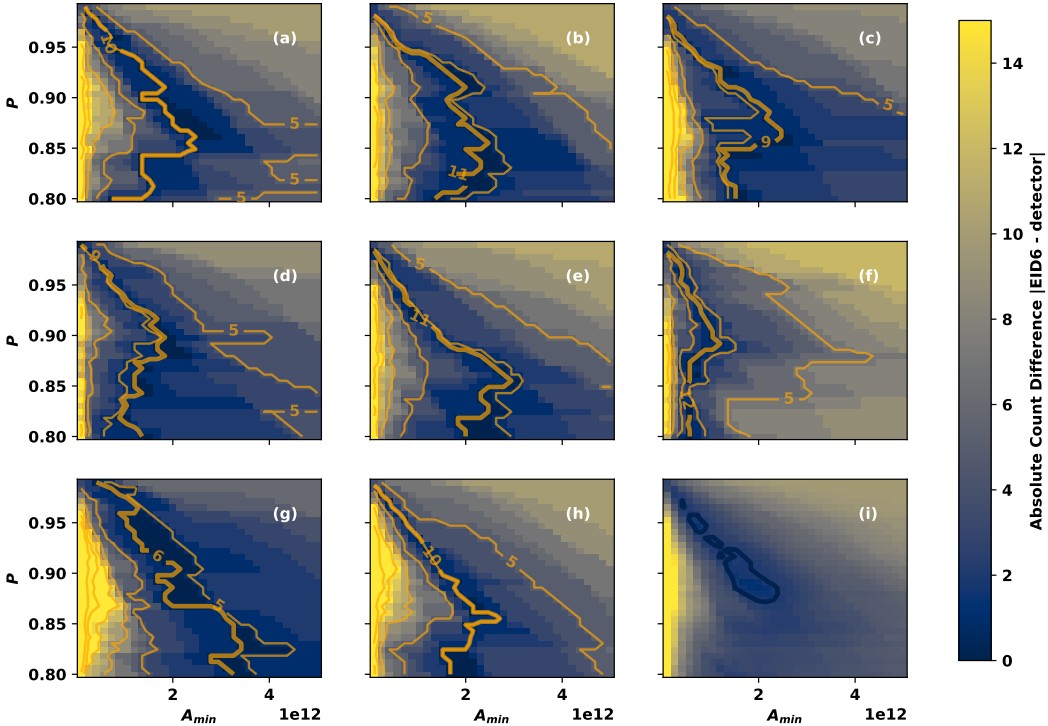

**Figure 6.** (a-h) Detected counts $N_i = F_3(P, A_{\min} | \Delta y = 15, \boldsymbol{Q_i})$ from eight random IVT fields $\boldsymbol{Q_i}$ (orange contours) as a function of $A_{\min}$ and $P$, with $\Delta y \approx 15$. Thin contours are drawn between 5 and 35 counts at intervals of 5. The bold orange contour shows the number of ARs counted by Expert ID 6. Shaded contours show the absolute difference between $F_3$ and the number of ARs counted by Expert ID 6. (i) The root-mean-squared average of the differences shown in (a–h). The bold blue contour shows the RMS difference of 2.

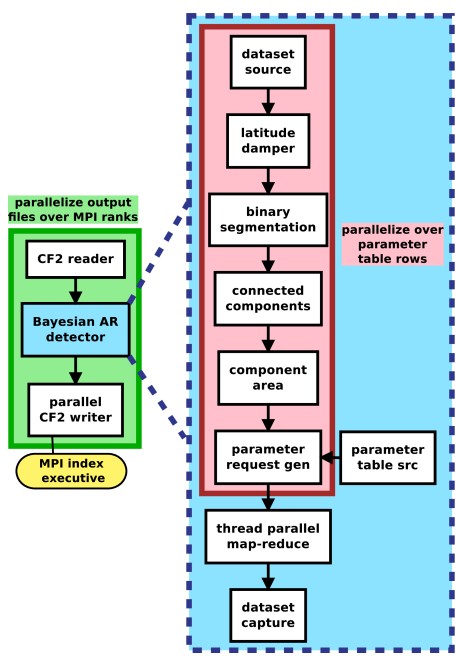

**Figure 7.** A diagram of the TECA pipeline that makes up the TECA Bayesian AR Detector application.

contour in Figures 6a–h to emphasize the parts of the parameter space that yield the same counts as the expert. Since we use a normal likelihood function (Equation 2), the log-posterior is proportional to $\sum_i (N_i - N_i')^2$. The shaded contours in Figures 6a–h illustrate the contribution of each field to the posterior distribution by showing $|N_i - N_i'|$ for the eight random IVT fields. Each field has a different portion of the $P$-$A_{\min}$ space where the difference between the detected counts and the expert counts are minimized. When these differences are combined–in a root-mean-squared sense–the result is an RMS difference field (Figure 6i) with multiple distinct minima: these minima translate to multiple distinct maxima in the EGID 6 posterior distribution. Similar reasoning applies to the multimodality in the posterior distributions associated with the other EGIDs.

One could interpret this multimodality as being a side-effect of having relatively few samples (133 in the case of EID 6); it is possible that having a higher number of samples would result in a smoother posterior distribution. It is also possible that the multimodality is associated with uncertainty in the expert counts themselves, such that under- or over-counting leads to distinct modes in the posterior distribution. This latter could possibly be dealt with by employing a more sophisticated Bayesian model: one that explicitly accounts for uncertainty in the expert data. Future work could explore such a possibility. Regardless, this analysis demonstrates that the multimodality is an inherent property of the detector-data system.

## 2.5 Implementation in the Toolkit for Extreme Climate Analysis

We implement the detector as an application in the Toolkit for Extreme Climate Analysis (TECA[2]). TECA is a framework for facilitating parallel analysis of petabyte-scale datasets. TECA provides generic modular components that implement parallel execution patterns and scalable I/O. These components can easily be composed into analysis pipelines that run efficiently at scale at high performance computing (HPC) centers. Figure 7 shows the modular components used to compose the TECA-BARD v1.0.1 application. TECA is primarily written in C++, and it offers Python bindings to facilitate prototyping of pipelines in a commonly-used scientific language. Early prototypes of TECA-BARD v1.0.1 were developed using these bindings, and the MCMC code invokes TECA-BARD via these Python bindings.

The TECA BARD v1.0.1 pipeline depicted in Figure 7 consists of a NetCDF reader (`CF2 reader`), the Bayesian AR Detector, and a NetCDF writer (`CF2 writer`). The Bayesian AR Detector component of the pipeline nests a separate pipeline consisting of the AR detection stages illustrated in Figure 3. The `thread parallel map-reduce` stage parallelizes the application of these AR detection stages over the 1,024 detector parameters (which are provided by `parameter table src` in combination with requests from `parameter request gen`) and passes on the reduced dataset to the `dataset capture` component, which passes that data on to the `CF2 writer`. The AR detection stages, for a given parameter set, are implemented as follows:

- `dataset source` takes IVT data from `CF2 reader` (Figure 3a),

- `latitude damper` uses the filter latitude width $\Delta y$ and applies Equation 3 (Figure 3b),

- `binary segmentation` identifies gridcells above the percentile threshold $P$ (Figure 3c),

- `connected components` finds contiguous regions where the percentile threshold is satisfied (Figure 3c), and

- `component area` calculates the areas of these contiguous regions and removes areas that are smaller than $A_{\min}$ (Figure 3d).

To improve performance on large calculations, TECA uses a map-reduce framework that takes advantage of both thread-level parallelism (using C++ threads) and multi-core parallelism (with the message passing interface MPI). TECA-BARD v1.0.1 distributes a range of MCMC parameters over different threads, and it distributes timesteps over different processes using MPI. This strategy allows TECA-BARD v1.0.1 to scale efficiently on HPC systems. We ran TECA-BARD v1.0.1, which effectively consists of 1,024 separate AR detectors, on 36.5 years of 3-hourly MERRA2 output (see Section 3) at the National Energy Research Scientific Computing center (NERSC) on the Cori system using 1,520 68-core Intel Xeon-Phi (Knights Landing) nodes in under 2 minutes (wallclock time).

---

[2]https://github.com/lbl-eesa/teca doi:10.20358/C8C651

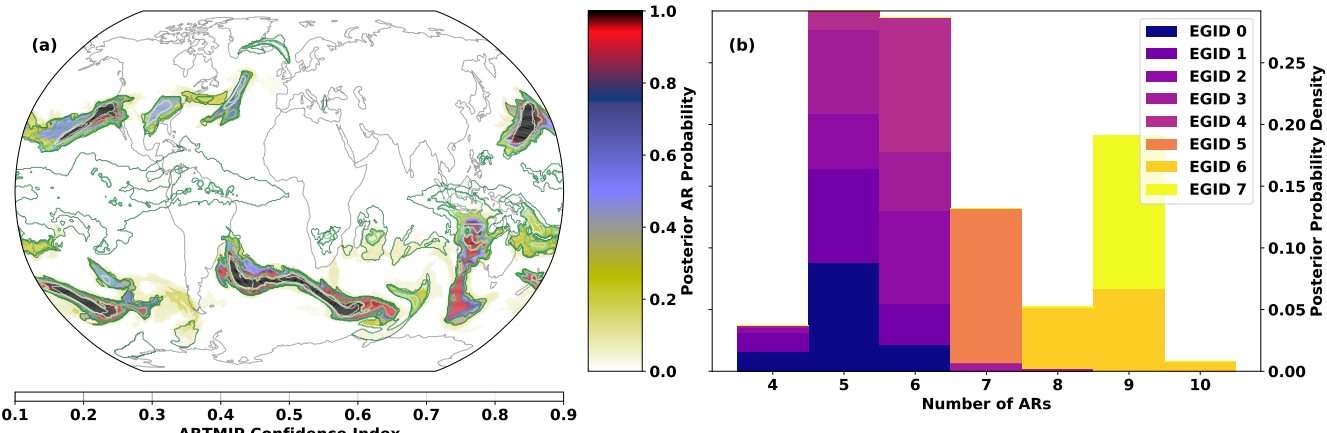

**Figure 8.** (a) AR detections on 7 February, 2019 at 0600Z. Filled contours show the posterior probability of AR detection $p_{AR}$ from TECA-BARD v1.0.1. Contour lines show the ARTMIP Confidence Index (the proportion of ARTMIP algorithms detecting an AR at a given location), $P_{ARTMIP}$. (b) Posterior distributions of counts for the same time slice, grouped by Expert Group ID (EGID).

## 3 Evaluation of TECA-BARD v1.0.1

We run TECA-BARD v1.0.1 on 3-hourly IVT output from the MERRA2 reanalysis (Gelaro et al., 2017) from 1 January, 1980 to 30 June 2017, which involves running the detector described in Section 2 for each of the 1,024 samples from the posterior distribution. Output from TECA-BARD v1.0.1 differs in character from other algorithm output in ARTMIP in that

it provides a posterior probability of AR detection $p_{AR}$, rather than a binary indicator of AR presence (Shields et al., 2018). We derive a comparable measure of AR presence by averaging binary AR identifications across available ARTMIP algorithms, on a location-by-location basis. This yields a probability-like quantity, which we refer to as the 'ARTMIP Confidence Index', $P_{ARTMIP}$: the proportion of ARTMIP algorithms reporting AR presence at each time slice. Output from TECA-BARD v1.0.1 is shown in Figure 8a, which also shows the corresponding ARTMIP Confidence Index for comparison.

TECA-BARD v1.0.1 and ARTMIP generally agree on the presence of 'high confidence' ARs: regions in which $p_{AR}$ and $P_{ARTMIP}$ are high. There are four regions of extremely high posterior AR probability in TECA-BARD-v1.0.1: areas where $p_{AR} \approx 1.0$ (regions with red and black coloring) in Figure 8a. All five of these regions are enclosed by white contours, indicating that at least 90% of ARTMIP algorithms also indicate AR presence. There are two additional distinct regions (in the the eastern U.S. and the central, north Atlantic) where $P_{ARTMIP} > 0.9$, whereas $p_{AR}$ only reaches approximately 0.6; these regions have

relatively small areas. Such behavior arises because of multimodality in the posterior distribution of parameters; e.g., Figure 5c shows that there are several distinct modes in the minimum area parameter. It is likely that these two regions of high IVT have areas that fall between two of these modes.

Most of the disagreement between ARTMIP and TECA-BARD v1.0.1 is associated with 'low confidence' AR regions: particularly regions in which the ARTMIP Confidence Index is in the range of 20%. The most prominent of these is a large region

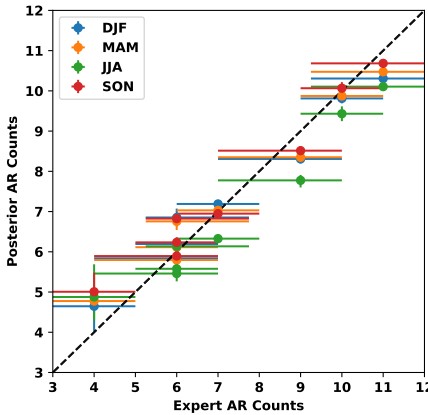

**Figure 9.** Posterior mean AR counts for each season, grouped by EGID vs median number of ARs counted by the corresponding Expert ID. There are four points (corresponding to the four seasons) for each EGID. Since Expert ID is assigned in order of increasing AR counts, the lowest EIDs occur on the left side of the graph and the highest occur on the right. Whiskers indicate the 5-95 percentile range. The dashed line shows the 1:1 line.

of $P_{\mathrm{ARTMIP}} \approx 0.2$ in the tropics; whereas $p_{\mathrm{AR}} \approx 0$ throughout the tropics. We argue that this represents erroneous detection of the ITCZ by a small subset of ARTMIP algorithms. The tropical filter (corresponding to parameter $\Delta y$) in TECA-BARD v1.0.1 explicitly filters out the tropics to avoid such erroneous detection of the ITCZ.

Figure 8b shows that TECA-BARD v1.0.1 detects 4–10 ARs in the dataset shown in Figure 8a. The range of uncertainty 5 is much smaller within individual Expert Group IDs (EGIDs); the most restrictive Expert Group ID (EGID 0) detects 4–6 ARs, whereas the most permissive parameter group (EGID 7) detects 9–10 ARs. This is consistent with the behavior shown in Figure2a; lower Expert IDs have lower counts and vice-versa.

More broadly, the number of ARs counted within each EGID is consistent with the number of ARs counted by the corresponding expert contributors. Figure 9 shows that the AR counts from the various EGIDs are consistent with AR count statistics 10 from the corresponding expert contributors. For all seasons, the points in Figure 9 are close–within error bars–to the one-to-one line. Note that we do not disaggregate expert counts by season, since doing so would lead to small sample sizes for some expert IDs. The seasonal range in posterior counts across EGIDs suggests that this should not affect our conclusion that EGIDs within TECA-BARD v1.0.1 emulate the counting statistics of corresponding experts, since the seasonal range is only approximately $\pm 1$.

15 Figure 9 also shows that the uncertainty in the number of detected ARs in TECA-BARD v1.0.1 is a direct consequence of uncertainty in the input dataset. Further, the spread in expert counts results in EGIDs having distinct groups of parameters. Figure 5 shows that the EGIDs associated with the most restrictive experts tend to have large minimum area parameters and narrower tropical filters, whereas the opposite is true for the most permissive EGIDs. This shows that the MCMC method yields a set of parameters that yield AR detectors that emulate the bulk counting statistics of the input data.

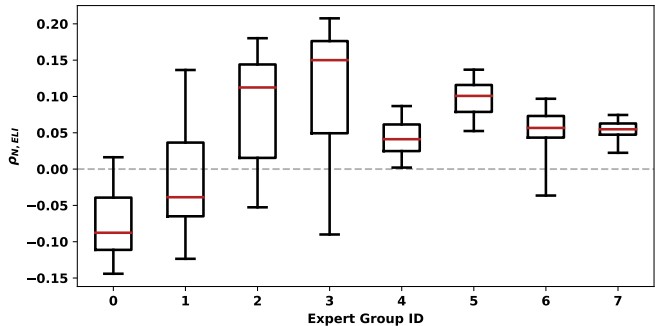

**Figure 10.** Box-and-whisker plots of the correlation between global AR count and ELI $\rho_{N,\mathrm{ELI}}$, grouped by EGID. Red lines show the median, box limits show the interquartile range, and whiskers indicate the 5-95% range.

## 4 Uncertainty in the Relationship between ENSO and AR Count

We assess the impact of parametric uncertainty in TECA-BARD v1.0.1 by asking a relatively simple question: *Are there more ARs during El Niño events?*. We examine this question from a global perspective, which is partly motivated by Guan and Waliser (2015b) (their Figures 10a,b), who show coherent changes in AR probability associated with the El Niño Southern

Oscillation (ENSO). The predominant effect is an equatorward shift of ARs during the positive phase of ENSO, and their figure seems to show more areas of increased AR occurrence than areas of decrease; this might suggest that positive phases of ENSO are associated with more ARs globally. Goldenson et al. (2018) indicate that, at least regionally, their analysis of the impact of ENSO on AR predictability leads to a different conclusion than that of Guan and Waliser (2015b). Goldenson et al. (2018) and Guan and Waliser (2015b) utilize different AR detection algorithms, which suggests that inferred relationships

between ENSO and ARs may depend on the detection algorithm used.

TECA-BARD v1.0.1 consists of 1,024 plausible AR detectors, which allows us to analyze whether there are significant differences in the answer to this question across the sets of AR detector parameters. We compare the TECA-BARD v1.0.1 output from MERRA2, described in Section 3, against the ENSO Longitude Index (ELI) of Williams and Patricola (2018). ELI represents the central longitude of areas in the tropical Pacific where sea surface temperatures are warmer than the zonal

mean, which–because of the weak temperature gradient approximation–is close to the longitude of maximum tropical Pacific convection. High values are associated with El Niño conditions and low values are associated with La Niña conditions. We calculate the average ELI for each boreal winter (December, January, and February) between 1981 and 2017. Similarly, we calculate the DJF-average number of detected ARs, over the same time period, for each of the 1,024 sets of parameters in TECA-BARD v1.0.1; we then calculate the Spearman rank correlation coefficient $\rho_{N,\mathrm{ELI}}$ between each set of DJF AR counts

and ELI. This yields 1,024 values of $\rho_{N,\mathrm{ELI}}$, which expresses the interannual correlation between DJF AR count and ELI for each set of AR detectors in TECA-BARD v1.0.1. Figure 10 shows the results of this calculation.

Across all EGIDs, the correlation coefficients range from approximately -0.2 to +0.2; they span zero. However, grouping results by EGID shows that different groups of detector parameters yield qualitatively different results. Figure 10 shows the

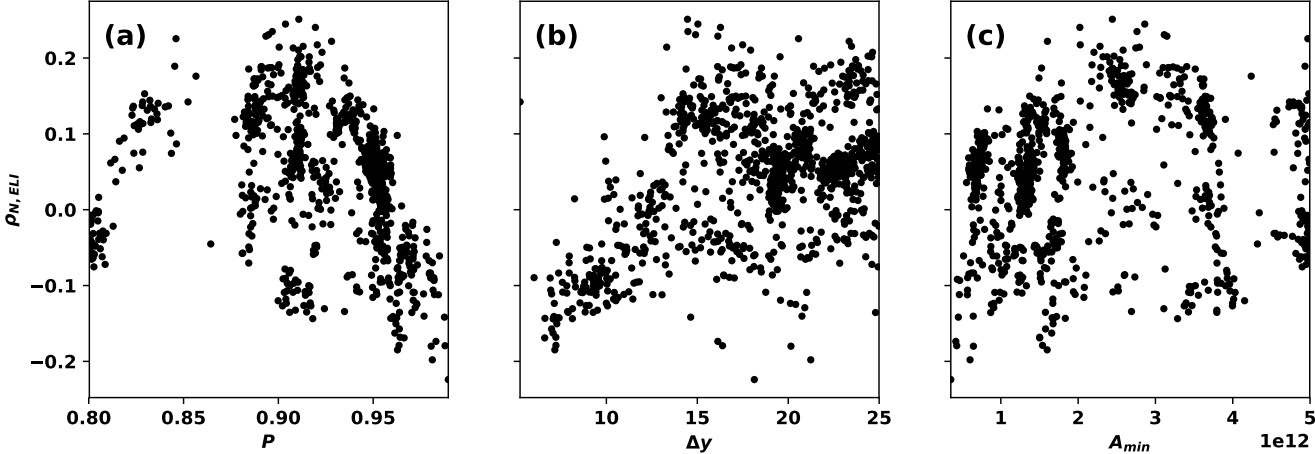

**Figure 11.** Correlation between global AR count and ELI $\rho_{N,\text{ELI}}$ as a function of AR detector parameter.

posterior statistics of $\rho_{N,\text{ELI}}$, grouped by EGID. EGIDs 0 and 1 have predominantly negative correlation coefficients (the medians and 5th percentile values are negative), though the 95th percentile values are positive. On the other hand, correlation coefficients from EGIDs 4, 5, and and 7 are all entirely positive, and most values from EGIDs 2, 3, and 6 are positive. Even within the uncertainty quantification framework of TECA-BARD v1.0.1, if we had utilized a single expert contributor–e.g.,
EGID 4, 5, or 7–we might have over-confidently concluded that there are more ARs globally during El Niño events.

It is intriguing that the most restrictive EGIDs tend to yield negative correlation coefficients, while the most permissive EGIDs tend to yield positive correlation coefficients. This variation appears to be controlled by variations in the percentile threshold $P$ and the tropical filter $\Delta y$. Figures 11a–c show samples of the posterior distribution of $\rho_{N,\text{ELI}}$ as a function of detector parameters. In Figures 11c, $\rho_{N,\text{ELI}}$ is evenly distributed across zero for the entire parameter space; whereas in Fig-
ures 11a,b the correlation coefficient show systematic variation with the input parameters $P$ and $\Delta y$. The largest values of $\Delta y$ and the smallest values of $P$ tend to be associated with positive values of $\rho_{N,\text{ELI}}$.

We further disaggregate results in Figure 12 by showing how $\rho_{N,\text{ELI}}$ clusters by EGID in two-dimensional projections of the parameter space. We utilize `fastKDE`[3] (O'Brien et al., 2016) to calculate two-dimensional marginal posterior distributions for each EGID: e.g., $p_j(P, \Delta y \mid N, Q)$ in Figure 12a (where $j$ corresponds to the EGID). We show contours of constant $p_j$,
colored by EGID, such that 95% of the posterior distribution for each EGID falls within the given contour; the colored contours in Figure 12 effectively outline the parameter samples for each EGID.

Parameter clusters with both positive $\rho_{N,\text{ELI}}$ and high $\Delta y$ tend to form distinct zones of points in Figure 12: clusters with relatively low $P$ and relatively high $\Delta y$. Parameters with negative $\rho_{N,\text{ELI}}$ predominantly fall along two lines in the $P$-$A_{\min}$ plane in Figure 12b, with the positive $\rho_{N,\text{ELI}}$ values forming on the line with lower $P$ values. These separate clusters are
associated with the more permissive EGIDs.

---

[3]https://bitbucket.org/lbl-cascade/fastkde at commit f2564d6

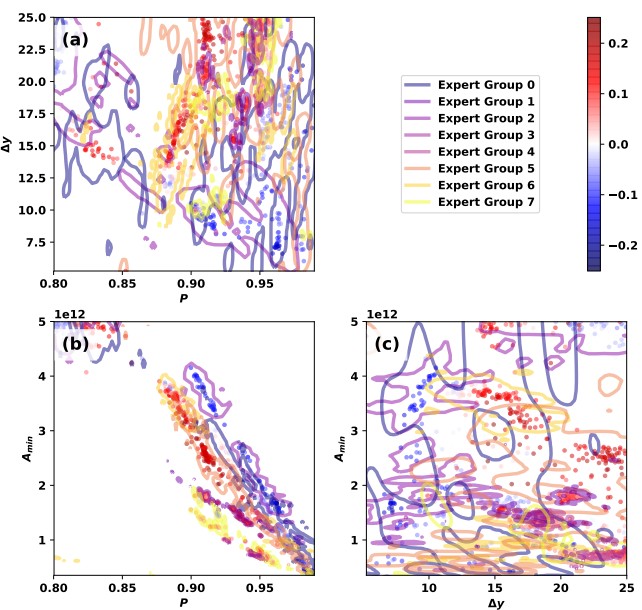

**Figure 12.** Pairplots of AR detector samples. Marker colors indicate the correlation between global AR count and ELI $\rho_{N,\text{ELI}}$, and colored lines show contours of constant $p_j$ such that 95% of the posterior distribution for each EGID falls within the given contour.

We argue that the differences in correlation coefficient between the restrictive and permissive EGIDs likely results from differences in the degree to which tropical moisture anomalies are filtered among the EGIDs. Patricola et al. (2019) show that strong El Niño events are associated with positive IVT anomalies in much of the tropics and a separate band of positive anomalies in the midlatitudes (around $30°$ latitude; their Figure 11). The positive IVT anomalies in the tropics would have no effect on the subset of AR detector parameters with high values of $\Delta y$, since these values would be aggressively filtered. This subset of parameters with high $\Delta y$–which is associated with the permissive EGIDs and positive values of $\rho_{N,\text{ELI}}$ (Figures 5b and 11b)–would then only be affected by the higher-than-average IVT in the midlatitudes. This would result in larger numbers of ARs during El Niño events. For AR detectors parameters with low values of $\Delta y$, the zone of positive anomaly in the tropics would not be totally filtered out, which increases the chances for zones of high IVT in the midlatitudes to be connected to zones of high IVT in the tropics. This could potentially result in larger-than-average, and fewer, ARs during El Niño.

## 5 The Importance of Uncertainty in Feature Detection

The results in Section 4 show that equally plausible sets of AR detector parameters can yield qualitatively different conclusions about the connection between ENSO and AR count. These results also show that the data used to constrain the AR detector parameters in TECA-BARD v1.0.1 has a huge influence on the choice of parameters and ultimately the conclusions that one might draw. Figure 10 shows that almost half of the spread in $\rho_{N,\text{ELI}}$ can be explained by the spread in expert counts used to

constrain the Bayesian model. This spread results from differences in subjective opinion about what does or does not constitute an AR.

There are numerous aspects of AR-related research for which TECA BARD v1.0.1 could be useful: including research on AR variability, predictability, and impacts in the observational record; and changes in AR dynamics and impacts in past and future climates. We use the ENSO-count relationship simply as a demonstration that parametric uncertainty can have a large effect on data analyses. There are numerous results in the literature for which a single AR detection method was used (or in some cases a few detection methods applied over multiple studies), including: 90% of the poleward moisture flux is associated with ARs (Zhu and Newell, 1998), 15–35% of precipitation in coastal California comes from ARs (Dettinger, 2011; Rutz et al., 2014; Guan and Waliser, 2015a; Gershunov et al., 2017; Rutz et al., 2019), there are 50–600% more AR days in RCP8.5 scenarios (Gao et al., 2015), RCP8.5 scenarios have two times more extreme precipitation associated with ARs in northern California (Gershunov et al., 2019a), etc. (Payne et al., 2020, and references therein). Many of the existing AR studies have considered uncertainty in the underlying datasets, such as uncertainty associated with choice of reanalysis and climate models (Gao et al., 2015; Payne and Magnusdottir, 2015; Warner et al., 2015; Espinoza et al., 2018; Gershunov et al., 2017, 2019a; Ralph et al., 2019b; Payne et al., 2020), and a few have considered AR detector uncertainty in the observational record of ARs (Guan and Waliser, 2015a; Ralph et al., 2019b). Studies based on ARTMIP have started to explore uncertainty with respect to AR detection, and the uncertainty is larger than many in the community had anticipated (Shields et al., 2018, 2019b; Chen et al., 2018; Rutz et al., 2019; Shields et al., 2019a; Chen et al., 2019; Ralph et al., 2019b; Payne et al., 2020). Preliminary results from the ARTMIP Tier 2 experiments suggest that AR detection uncertainty may be comparable to model uncertainty in future climate simulations (O'Brien et al., 2020), which implies that ongoing AR research would benefit from consideration of AR detection uncertainty. TECA BARD v1.0.1 offers an efficient way for future studies to quantify AR detection uncertainty in-situ.

In the current literature, AR detectors have two main developmental stages: (1) decide on the steps used in the AR detection algorithm, and (2) determine values used for unconstrained parameters (e.g., thresholds like $P$, $\Delta y$, and $A_{\min}$). In all examples of AR literature known to these authors, both steps rely on expert judgement. If we frame this in terms of the AR detector described in Section 2.3, step (2) would involve an expert varying the detector parameters $P$, $\Delta y$, and $A_{\min}$ until the resulting AR detections are acceptable. It seems reasonable to assume that if Expert ID 0 were to manually choose parameters in such a way, they would likely choose parameters that would yield a negative correlation between ENSO and global AR count; conversely, Expert ID 7 would almost certainly choose parameters that would yield a positive correlation coefficient. Setting aside uncertainty in the detector design (stage 1), two different experts could potentially develop AR detectors that would come to opposite conclusions about the impact of ENSO on AR count.

It is crucial to recognize the importance and impact of this spread in subjective opinion. Subjective opinion is currently used in the literature to define quantitative methods for detecting ARs. Since we currently lack physical theories to constrain AR detection schemes like this, such as theories about what the number of ARs should be, subjective opinion is the only option. These results show that subjective opinion can qualitatively impact the conclusions that one might draw. It therefore seems imperative to reduce uncertainty, though it is not immediately clear how that might be achieved. Adding more walkers to the

MCMC calculation described in Section 2.4 would not change the underlying posterior distribution; it would only sample it more thoroughly, which would somewhat increase the spread in parameters. Adding more expert contributors (and possibly more contributions from each contributor) could have one of two main outcomes: (1) if a consensus were to emerge about AR counts, then it is possible that the EGID posterior distributions $p_j$ would start to form a 'consensus' in the combined posterior distribution, with reduced spread in the parameter space; or (2) it is possible that each new expert contribution results in a new mode appearing in the parameter space, such that uncertainty is actually increased by adding more expert contributions. Moreover, it is not clear whether the reduced parameter spread associated with outcome (1) would be desirable, since it would weight the parameter selection toward the 'consensus' of EGIDs, at the expense of suppressing 'outlier' EGIDs. The answer to this question is somewhat philosophical in nature, and the answer is likely to be application-dependent. Ultimately, physical theories about ARs may be the only reasonable way to constrain AR detection methods and therefore reduce uncertainty associated with subjective opinion.

This study considers the parametric uncertainty in a single detector framework, and it does not consider the structural uncertainty in the detector framework itself. This is a key limitation of this study, and it is an opportunity for expanding this work in future studies. For example, we could have utilized an absolute threshold in IVT (e.g., 250 kg m$^{-1}$ s$^{-1}$) rather than a relative, percentile-based threshold. One might imagine applying the general Bayesian framework described in Section 2.1 to other existing AR detectors in the literature as a way to explore both structural and parametric uncertainty. The expert count data produced as part of this study, which are publicly available following information in the *Code and data availability* statement at the end of this manuscript, could readily be used for such an exercise.

We base TECA BARD v1.0.1 on input from 8 experts who co-authored this study (see *Author contributions* at the end), which may limit the range of uncertainty that TECA BARD v1.0.1 can explore. If there is sampling bias in the expert counts, it is also possible that use of a limited sample size could bias the detector toward a particular definition of AR. Figure 12 shows that each EGID results in parameters that are grouped somewhat closely together in parameter space, so it is reasonable to assume that additional experts would result in new EGIDs with different groupings of parameters. There are two main reasons that we limit this study to contributions from only 8 experts: the amount of person-effort required to solicit input, and the computational expense of training the Bayesian model on each expert. In addition to the substantial person-effort invested by each additional contributor, engaging more experts would require soliciting input from experts outside of the project that funded this effort (see the *Acknowledgements* section), which would require investing in further development of the GUI (Figure 1) to port it to other systems. It seemed prudent to limit our investments in such further developments, since our initial data collection phase concluded right about the same time that the ClimateNet effort (see two paragraphs down) launched.

One could consider utilizing data from the ARTMIP project to constrain a Bayesian model, since each ARTMIP catalogue effectively represents each expert developer's opinion on where and when ARs can be distinguished from the background. This would greatly increase the effective number of experts, though it would likely also require a substantially more complicated Bayesian model. As noted by Ralph et al. (2019b), each existing AR detection algorithm has been designed for a specific application: ranging from understanding the global hydrological cycle (Zhu and Newell, 1998) to understanding AR impacts in the western U.S. (Rutz et al., 2014). Forthcoming work by Zhou et al. (2020) shows that the global number of ARs detected

by ARTMIP algorithms ranges from approximately 6 to 42. This is a much wider range of uncertainty in global AR count than demonstrated in this manuscript, and we hypothesize that the large upper bound is a side-effect–rather than an intended property–resulting from designing AR detectors with a focus on a particular region or impact. For example, if an AR detector designer is not particularly concerned about ARs being strictly contiguous, then global AR count would not be well constrained.

If global AR count is not a reliable reflection of the AR detector designer's expert opinion, then we would need to either account for this uncertainty in the ARTMIP dataset, or we would need to formulate likelihood functions that optimize based on some other property of the ARTMIP output: ideally, properties that reflect expert opinion.

The use of counts, instead of AR footprints, is potentially another limitation of this study that could be explored in future work. For example, during the MCMC training phase, some parameter choices may yield some (false positive) detections of
tropical cyclones; these false positives are not penalized, since a likelihood function based entirely on counts has no way of discriminating between true and false positives. We could employ additional heuristic rules to filter out common false positives like tropical cyclones (e.g., by filtering out ARs in which $\nabla \times \overrightarrow{\text{IVT}}$ exceeds a threshold). Alternatively, using AR footprints in the training phase could help narrow the parameter choices to ones that minimize such false positives; however, the availability and quality of such data could be a concern. Prabhat et al. (2020) have created a web interface for soliciting user opinions about
the boundaries of ARs and tropical cyclones, which may be a more informative dataset for constraining an AR detector: they call this dataset *ClimateNet*. Prabhat et al. (2020) train a deep neural network to emulate the hand-drawn AR labels, and they show that this approach is broadly successful. The Bayesian approach described in this manuscript can be viewed as a form of statistical machine learning: training a heuristic detector to emulate the behavior of experts. The Bayesian approach could alternatively be tailored to utilize data from ClimateNet instead of–or in addition to–the count dataset used here. For example,
the posterior distribution of AR detector parameters could be used as a prior distribution for parameters in a model that uses some measure of *closeness* between the detected ARs and the ClimateNet ARs: e.g., the likelihood could be based around the intersection-over-union metric that is commonly applied in the computer vision literature. There are a number of interesting hypotheses, related to the TECA BARD approach, that could be explored in future studies:

– Hypothesis 1: ClimateNet provides a more information-rich dataset for constraining detector parameters, which could
25        be critical for reducing the parametric uncertainty shown in this study.

– Hypothesis 2: The spread in subjective opinion about what does and does not constitute an AR is large enough that the parametric uncertainty cannot be reduced further than that shown in this study.

– Hypothesis 3: Deep learning methods can outperform the statistical machine learning approach employed here.

– Hypothesis 4: The output from TECA-BARD v1.0.1 could be used to pre-train a Deep Learning model, so that it can
30        make better use of the spatial data in ClimateNet

The TECA BARD approach could also be applied to detectors of other types of weather phenomena. For example, the U.S. Clivar Hurricane Working Group determined that some tropical cyclone research results depend on how tropical cyclones are detected: particularly results concerning weaker cyclones (Walsh et al., 2015). Similarly, the Intercomparison of Mid

Latitude Storm Diagnostics (IMILAST) project determined that scientific results regarding extratropical cyclones can depend on how they are detected (Neu et al., 2013). There is also emerging research on frontal systems that could be interpreted to suggest a similar uncertainty with respect to tracking method (Schemm et al., 2018). We argue that such uncertainty is inherent to heuristic phenomena detectors, and Bayesian approaches like the one described in Section 2.1 could be used to quantify this
uncertainty.

*Code and data availability.*

The code for TECA-BARD v1.0.1 is available at https://github.com/LBL-EESA/TECA under TECA release 'TECA-BARD-v1.0.1'. TECA-BARD v1.0.1 is available as a TECA application `teca_ar_detect` under source file `apps/teca_bayesian_ar_de` (compiles as `bin/teca_bayesian_ar_detect` when installed). The code for sampling the posterior distribution of the
TECA-BARD parameters is available at https://bitbucket.org/lbl-cascade/bayesian_ar_detector. Data containing the AR counts used for constraining the TECA-BARD parameters are available at https://portal.nersc.gov/archive/home/projects/cascade/www/teca_bard/ar_count_files.tar and the posterior distributions samples are available under the TECA source file `alg/teca_bayesian`

*Author contributions.*  TAO was responsible for development of the concept, development of the statistical method, implementation of method, generation of figures, and initial drafting of the manuscript. MDR contributed to the development of the statistical method. TAO,
BL, AAE, HK, JNJ, and Prabhat all contributed to the implementation of the method in TECA. TAO, CMP, JPO, AM, SAR, AMR, AC, and HAI contributed to the database of AR counts. All authors contributed to the editing of the manuscript.

*Competing interests.*  The authors declare they have no conflict of interest.

*Acknowledgements.*  This research was supported by the Director, Office of Science, Office of Biological and Environmental Research of the U.S. Department of Energy Regional and Global Climate Modeling Program (RGCM) and used resources of the National Energy Research
Scientific Computing Center (NERSC), also supported by the Office of Science of the U.S. Department of Energy under Contract No. DE-AC02-05CH11231. The authors thank Dr. Christopher J. Paciorek for providing useful input on the manuscript. The authors would like to express their sincere gratitude for input from two anonymous reviewers, whose comments greatly improved the presentation of the methodology and the resulting discussion.

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
