# Peer review of "Detection of Atmospheric Rivers with Inline Uncertainty Quantification: TECA-BARD v1.0.1"

_Geoscientific Model Development, 2020_

## Referee Comment (RC1) · Anonymous Referee #1 · 13 May 2020

General comments:

O'Brien et al. document a process for detecting atmospheric rivers (ARs) using expert input combined with statistical tools. They also include some uncertainty quantification in the form of parametric uncertainty associated with the correlation between ARs and ENSO events. This is an important topic of research, the study is well designed, and the paper is well written. There are a few places that could use additional context/discussion, and I have highlighted these below. Once these and other comments are addressed, I recommend the paper for publication in GMD.

1) Here the authors use a dataset of AR counts instead of a dataset of AR footprints

to develop the framework. This factor is mentioned briefly in the introduction, but worth more discussion on potential limitations/caveats.

2) Number of experts/samples: Is 8 experts enough? There is clearly a lot of variation among them. I found it fascinating that the authors are able to quantify the spread in expert judgement regarding how to define ARs (e.g., Figure 8). I would like to see more discussion on the "traits" of the expert groups and how they impact the results.

3) The multimodality of parameter posterior distributions could use more discussion (Fig 4). Can these be interpreted in a meaningful way?

4) What other applications (besides ENSO/AR relationship) might be worthwhile to test parametric uncertainty?

Specific comments:

Page 2, Line 29-30: I believe the ClimateNet project is working to achieve this type of dataset (spatial footprints of ARs). It is worth mentioning here and/or in the discussion how that effort might contribute to this topic and related future work.

Page 3, Line 26: Does the user see IVT only, or are there multiple fields available to diagnose ARs?

Page 7, Line 2: What does $\sigma$ represent?

Section 2.3.1: Is there some way to visualize the content in this section (like Fig 3 for the previous section)? It can be challenging to follow equations only.

Section 2.3.1: Why does the f equation on page 7 have sin(delta y) but equation 5 on page 8 has sin(2*delta y)?

Page 8, Line 1: How does grid cell size (i.e., resolution) impact these constraints?

Page 8, Line 23: How was statistical equilibrium determined? Autocorrelation time?

Page 9, Line 6ff: Some clarification is needed on the difference between Expert ID and

[Figure]

Expert Group ID.

Page 9, Line 7: How were 128 samples acquired for each Expert ID, when some experts did not contribute that many? Unless I am misinterpreting the numbers reported in Figure 2a and line 7 on page 4 ("between 64 and 906 time slices").

Figure 7: Where is Expert Group ID shown in this figure? Caption could use some clarification.

Page 13, Line 5: Please explain what the ELI is; what does it physically represent?

Technical corrections:

Figure 1: Recommend increasing the size of this image. It is difficult to see the green X's, for example.

Page 6, Line 6: Lower bound of parameter P is listed in text as 0.999 but in Table 1 as 0.99.

Section 2.3.1: Some of the equations don't have numbers (e.g., equations for A and f).

Page 7, Line 24: I think subscript of N_t should be capital (N_T), unless this is a different quantity.

Figure 5: Recommend increasing the size of this image to make text more readable.

Figure 10: Expert Group colored lines could be bolder/brighter (hard to see on panels, especially yellow colors).

---

## Referee Comment (RC2) · Anonymous Referee #2 · 13 May 2020

Summary of remarks:

The key point of this manuscript is that TECA-BARD v1.0 can be used to quantify uncertainty associated with parameter selection in atmospheric river detection models. To this end, the manuscript accomplishes its objective. I do have some questions regarding the number of experts' opinions used in the creation of TECA-BARD v1.0 and whether it is also feasible to use the output from other AR tracking algorithms to inform the posterior. Regardless, I think this manuscript is novel in its method for quantifying uncertainty associated with atmospheric river tracking algorithms, and deserves to be published once the following comments are addressed.

[Figure]

Major remarks:

1. The authors' basis for the paper is that the best dataset on which to base an atmospheric river tracking algorithm would include the outlines or counts from multiple domain experts. The authors also state that there is no quantitative definition of atmospheric rivers, and that only recently was a qualitative definition created. In response to these issues, the authors use eight expert opinions on the counts of atmospheric rivers to constrain the posterior of their Bayesian model.

However, there could also be issues with a collection of subjective opinions from domain experts being used, without background into how each expert views the quantitative and/or qualitative definition of an atmospheric river. It would be useful for this reason to have many more than 8 experts (an admittedly challenging prospect). Otherwise, there is potential for the dataset to be biased towards one type of atmospheric river definition through sampling bias of expert opinions.

2. Another possible Bayesian model for atmospheric river detection could instead use the various algorithms presented within ARTMIP to constrain the posterior. These algorithms required the combined efforts of more scientists than were used to provide the atmospheric river counts for the current study. It would therefore be helpful for the authors to address this alternative, and to discuss the implications of using a collection of experts' opinions rather than a collection of previously generated algorithms.

3. The large uncertainty that remains within TECA-BARD v1.0 even with the usage of 1,024 unique AR detectors further suggests that including more experts and/or MCMC chains per expert would help converge on a more statistically robust atmospheric river detection method. The qualitative conclusion regarding the uncertainty of the sign of correlation between ENSO phase and atmospheric river frequency is valid regardless of the remaining uncertainty within TECA-BARD v1.0. However, if TECA-BARD is to be used for more quantitative assessments of relationships between atmospheric rivers and climate modes (ENSO, PDO, etc.), it would be useful to constrain the uncertainty

of the model even further.

Minor remarks:

1. It would be useful to cite and discuss recent efforts to collect domain-expert defined outlines of atmospheric rivers, as discussed in Prabhat et al. (2020); https://www.geosci-model-dev-discuss.net/gmd-2020-72/

2. Typo on line 20, page 13: Change "Figures 9" to "Figure 9"

3. Could the parameter space of Figure 10 be filled by using a greater number of samples from each expert? It may be illustrative to fill more of this parameter space, although I understand issues with computational limitations.

4. The posterior PDFs for the combined EGID model shown in Figure 4 exhibit multi-modality. Do you think this multi-modality is caused by the limited number of experts used to constrain the posterior, or is there some other cause? Either way, it would be useful to discuss this within the manuscript.

5. I recommend increasing the size of Figure 5 so it is easier to read in the final manuscript. Also, it could be useful to reference the various phrases used in the flowchart within the manuscript itself.

---

## Author Response (AR1)

We thank both reviewers for their constructive comments, which have improved the clarity and presentation of the paper. We respond to each reviewer comment below, with the following format:

> The quoted reviewer comment appears as unindented, red text with a vertical bar to the side of the quoted text.
>
> Our responses show as blue, indented text.

(orig. pA.B, cur. pC.D): "And non-trivial changes to the text are quoted in gray. The original page 'A' and line number 'B' of any corresponding change, as well as the page number 'C' and new line number 'D' of the change in the current manuscript version, are encoded at the beginning of any quoted change as '(orig. pA.B, cur. pC.D)'."

We discuss changes related to re-running the MCMC code in the first section, we address the major comments from reviewer 1 and reviewer 2 in the second two sections. We then address the specific/minor comments from each reviewer in the final section.

**Re-running the MCMC Sampling**

In responding to one of Reviewer 1's comments–about the appearance of a factor of 2 in Equation 5, which describes the geometric constraint on the prior–we discovered that our prior code also had the erroneous value of 2, which made the prior overly restrictive: particluarly with respect to values of $\Delta y$. We also discoevered that our MCMC code was not properly accounting for TECA-BARD reporting the background field in the list of connected components, which offset all of the reported counts by +1. We fixed both of these issues, re-ran the MCMC code, re-ran all subsequent analyses, and regenerated all figures that depend on the output of TECA-BARD.

These changes resulted in no changes to the main point of the manuscript, and they resulted in few qualitative changes. We note here changes that were made as a result of repeating the MCMC sampling and all subsequent data analysis steps:

- updated the algorithm version in the title and throughout the manuscript to 1.0.1

- Figures 4 and 6–10 (numbering relative to the original submission) were updated

- changed the number of walkers for each EGID from 1,024 to 128, in order to reduce the computational cost of re-running the MCMC calculation.

- made minor modifications to several areas of the text to bring the description in the text in line with the new figures (e.g., description of the posterior counts in the comparison with ARTMIP output)

- made a significant change to one paragraph in Section 4:
  (orig. p14.6, cur. p18.17): "Parameter clusters with both positive $\rho_{N,\mathrm{ELI}}$ and high $\Delta y$ tend to form distinct zones of points in Figure 12: clusters with relatively low $P$ and relatively high $\Delta y$. Parameters with negative $\rho_{N,\mathrm{ELI}}$ predominantly fall along two lines in the $P$-$A_{\min}$ plane in Figure 12b, with the positive $\rho_{N,\mathrm{ELI}}$ values forming on the line with lower $P$ values. These separate clusters are associated with the more permissive EGIDs."

**Reviewer 1 Major Comments**

1) Here the authors use a dataset of AR counts instead of a dataset of AR footprints to develop the framework. This factor is mentioned briefly in the introduction, but worth more discussion on potential limitations/caveats.

We agree, especially in light of the new availability of such a dataset. We combine our response to this comment with a response to comments echoed by both reviewers that Prabhat et al. (2020) should be discussed. We added some discussion of Prabhat et al. (2020) to the introduction, and we have added substantial discussion of the limitations of using a count-based dataset and future possibilities for utilizing ClimateNet.

**(orig. p2.29, cur. p2.31):** "The best type of dataset would presumably be one in which experts outline the spatial footprints of ARs, such as the ClimateNet dataset described in the forthcoming manuscript by Prabhat et al. (2020). At the time that the work on this manuscript started, the ClimateNet dataset did not yet exist, and we considered that the simpler alternative would be to identify the number of ARs in a set of given meteorological fields. Even though a dataset of AR counts is perhaps less informative than a dataset of AR footprints, we hypothesize that such a dataset could serve to constrain the parameters in a given AR detector."

**(orig. p16.12, cur. p22.8):** " The use of counts, instead of AR footprints, is potentially another limitation of this study that could be explored in future work. For example, during the MCMC training phase, some parameter choices may yield some (false positive) detections of tropical cyclones; these false positives are not penalized, since a likelihood function based entirely on counts has no way of discriminating between true and false positives. We could employ additional heuristic rules to filter out common false positives like tropical cyclones (e.g., by filtering out ARs in which $\nabla \times \overrightarrow{\text{IVT}}$ exceeds a threshold). Alternatively, using AR footprints in the training phase could help narrow the parameter choices to ones that minimize such false positives; however, the availability and quality of such data could be a concern. Prabhat et al. (2020) have created a web interface for soliciting user opinions about the boundaries of ARs and tropical cyclones, which may be a more informative dataset for constraining an AR detector: they call this dataset *ClimateNet*. Prabhat et al. (2020) train a deep neural network to emulate the hand-drawn AR labels, and they show that this approach is broadly successful. The Bayesian approach described in this manuscript can be viewed as a form of statistical machine learning: training a heuristic detector to emulate the behavior of experts. The Bayesian approach could alternatively be tailored to utilize data from ClimateNet instead of–or in addition to–the count dataset used here. For example, the posterior distribution of AR detector parameters could be used as a prior distribution for parameters in a model that uses some measure of *closeness* between the detected ARs and the ClimateNet ARs: e.g., the likelihood could be based around the intersection-over-union metric that is commonly applied in the computer vision literature. There are a number of interesting hypotheses, related to the TECA BARD approach, that could be explored in future studies:

- Hypothesis 1: ClimateNet provides a more information-rich dataset for constraining detector parameters, which could be critical for reducing the parametric uncertainty shown in this study.

- Hypothesis 2: The spread in subjective opinion about what does and does not constitute an AR is large enough that the parametric uncertainty cannot be reduced further than that shown in this study.

- Hypothesis 3: Deep learning methods can outperform the statistical machine learning approach employed here.

- Hypothesis 4: The output from TECA-BARD v1.0.1 could be used to pre-train a Deep Learning model, so that it can make better use of the spatial data in ClimateNet

"

2) Number of experts/samples: Is 8 experts enough? There is clearly a lot of variation among them. I found it fascinating that the authors are able to quantify the spread in expert judgement regarding how to define ARs (e.g., Figure 8). I would like to see more discussion on the "traits" of the expert groups and how they impact the results.

We have added some discussion regarding this point.

**(orig. p16.12, cur. p21.19):** "We base TECA BARD v1.0.1 on input from 8 experts who co-authored this study (see *Author contributions* at the end), which may limit the range of uncertainty that TECA BARD v1.0.1 can explore. If there is sampling bias in the expert counts, it is also possible that use of a limited sample size could bias the detector toward a particular definition of atmospheric river. Figure 12 shows that each EGID results in parameters that are grouped somewhat closely together in parameter space, so it is reasonable to assume that additional experts would result in new EGIDs with different groupings of parameters. There are two main reasons that we limit this study to contributions from only 8 experts: the amount of person-effort required to solicit input, and the computational expense of training the Bayesian model on each expert. In addition to the substantial person-effort invested by each additional contributor, engaging more experts would require soliciting input from experts outside of the project that funded this effort (see the *Acknowledgements* section), which would require investing in further development of the GUI (Figure 1) to port it to other systems. It seemed prudent to limit our investments in such further developments, since our initial data collection phase concluded right about the same time that the ClimateNet effort (see next paragraph) launched."

3) The multimodality of parameter posterior distributions could use more discussion (Fig 4). Can these be interpreted in a meaningful way?

We have expanded the discussion in Section 2.4.1 to address this; we have also added a new figure (Figure 6 in the revised manuscript, Figure 1 in this document) to support the discussion.

**(orig. p9.10, cur. p11.17):** " The posterior distributions exhibit multimodality: both in the individual EGID posterior distributions and in the combined posterior distributions shown in Figure 6. This multimodality arises as a consequence of three factors: (1) parameter-dependence of the counts generated by the AR detector, which depends on the underlying IVT field being analyzed, (2) variability in the counts from each expert, and (3) the addition of posterior distributions from each EGID–each having their own distinct modes. To illustrate how the first two factors lead to inherent multimodality, Figures 6a–h show the dependence of the counts generated by the AR detector on the percentile and minimum area thresholds (orange contours): $F_3(P, A_{\min}|\Delta y = 15, \boldsymbol{Q_i})$ for eight random IVT fields $\boldsymbol{Q_i}$. $F_3$ exhibits similar qualitative dependence on $P$ and $A_{\min}$ for all eight cases: AR count tends to be high for low values of both $P$ and $A_{\min}$, and it tends to be low when both $P$ and $A_{\min}$ are high (see Section 2.3.1 for an explanation of the geometric relationship that leads to this behavior). Aside from this general qualitative agreement, the fine-scale details of the dependence of $F_3$ on $P$ and $A_{\min}$ depends strongly on the actual IVT field (compare Figures 6 (a) and (f) for example). Non-monotonic dependence of $F_3$ on the input parameters arises, for example, from ARs merging as $P$ is reduced or splitting as $P$ is increased (merging reduces the count, splitting increases the count). It is not surprising that the number of ARs detected depends simultaneously on the parameters controlling the AR detector and the IVT field in which ARs are being detected. The number of ARs counted by a given expert also depends on the given IVT field. The bold orange contour in Figures 6a–h shows the number of ARs counted by Expert ID 6; $N_i'$ is

a single scalar number for each field $Q_i$, and we show it as a contour in Figures 6a–h to emphasize the parts of the parameter space that yield the same counts as the expert. Since we use a normal likelihood function (Equation 2), the log-posterior is proportional to $\sum_i (N_i - N_i')^2$. The shaded contours in Figures 6a–h illustrate the contribution of each field to the posterior distribution by showing $|N_i - N_i'|$ for the eight random IVT fields. Each field has a different portion of the $P$-$A_{\min}$ space where the difference between the detected counts and the expert counts are minimized. When these differences are combined–in a root-mean-squared sense–the result is an RMS difference field (Figure 6i) with multiple distinct minima: these minima translate to multiple distinct maxima in the EGID 6 posterior distribution. Similar reasoning applies to the multimodality in the posterior distributions associated with the other EGIDs. One could interpret this multimodality as being a side-effect of having relatively few samples (133 in the case of EID 6); it is possible that having a higher number of samples would result in a smoother posterior distribution. It is also possible that the multimodality is associated with uncertainty in the expert counts themselves, such that under- or over-counting leads to distinct modes in the posterior distribution. This latter could possibly be dealt with by employing a more sophisticated Bayesian model: one that explicitly accounts for uncertainty in the expert data. Future work could explore such a possibility. Regardless, this analysis demonstrates that the multimodality is an inherent property of the detector-data system. "

4) What other applications (besides ENSO/AR relationship) might be worthwhile to test parametric uncertainty?

This was quite the oversight to neglect adding text suggesting possible other uses for TECA BARD: we thank the reviewer for pointing out this gap! We have added a paragraph and numerous references to Section 5.

(orig. p15.8, cur. p20.3): "There are numerous aspects of AR-related research for which TECA BARD v1.0.1 could be useful: including research on AR variability, predictability, and impacts in the observational record; and changes in AR dynamics and impacts in past and future climates. We use the ENSO-count relationship simply as a demonstration that parametric uncertainty can have a large effect on data analyses. There are numerous results in the literature for which a single AR detection method was used (or in some cases a few detection methods applied over multiple studies), including: 90% of the poleward moisture flux is associated with atmospheric rivers (Zhu and Newell, 1998), 15–35% of precipitation in coastal California comes from ARs (Dettinger, 2011; Rutz et al., 2014; Guan and Waliser, 2015a; Gershunov et al., 2017; Rutz et al., 2019), there are 50–600% more AR days in RCP8.5 scenarios (Gao et al., 2015), RCP8.5 scenarios have two times more extreme precipitation associated with ARs in northern California (Gershunov et al., 2019a), etc. (Payne et al., 2020, , and references therein). Many of the existing AR studies have considered uncertainty in the underlying datasets, such as uncertainty associated with choice of reanalysis and climate models (Gao et al., 2015; Payne and Magnusdottir, 2015; Warner et al., 2015; Espinoza et al., 2018; Gershunov et al., 2017, 2019a; Ralph et al., 2019b; Payne et al., 2020), and a few have considered AR detector uncertainty in the observational record of ARs (Guan and Waliser, 2015a; Ralph et al., 2019b). Studies based on ARTMIP have started to explore uncertainty with respect to AR detection, and the uncertainty is larger than many in the community had anticipated (Shields et al., 2018, 2019b; Chen et al., 2018; Rutz et al., 2019; Shields et al., 2019a; Chen et al., 2019; Ralph et al., 2019b; Payne et al., 2020) Preliminary results from the ARTMIP Tier 2 experiments suggest that AR detection uncertainty may be comparable to model uncertainty in future climate simulations (OBrien et al., 2020), which implies that ongoing AR research would benefit from consideration of AR detection uncertainty. TECA BARD v1.0.1 offers an efficient way for future studies to quantify AR detection uncertainty in-situ."

[Figure]

Figure 1: (a-h) Detected counts $N_i = F_3(P, A_{\min}|\Delta y = 15, \boldsymbol{Q_i})$ from eight random IVT fields $\boldsymbol{Q_i}$ (orange contours) as a function of $A_{\min}$ and $P$, with $\Delta y \approx 15$. Thin contours are drawn between 5 and 35 counts at intervals of 5. The bold orange contour shows the number of ARs counted by Expert ID 6. Shaded contours show the absolute difference between $F_3$ and the number of ARs counted by Expert ID 6. (i) The root-mean-squared average of the differences shown in (a–h). The bold blue contour shows the RMS difference of 2.

**Reviewer 2 Major Comments**

1. The authors' basis for the paper is that the best dataset on which to base an atmospheric river tracking algorithm would include the outlines or counts from multiple domain experts. The authors also state that there is no quantitative definition of atmospheric rivers, and that only recently was a qualitative definition created. In response to these issues, the authors use eight expert opinions on the counts of atmospheric rivers to constrain the posterior of their Bayesian model.

However, there could also be issues with a collection of subjective opinions from domain experts being used, without background into how each expert views the quantitative and/or qualitative definition of an atmospheric river. It would be useful for this reason to have many more than 8 experts (an admittedly challenging prospect). Otherwise, there is potential for the dataset to be biased towards one type of atmospheric river definition through sampling bias of expert opinions.

We agree that this is an important limitation of this study. We have added text explaining why this study uses 8 experts and we have added text to the discussion to make this limitation more clear.

(orig. p16.12, cur. p21.19): "We base TECA BARD v1.0.1 on input from 8 experts who co-authored this study (see *Author contributions* at the end), which may limit the range of uncertainty that TECA BARD v1.0.1 can explore. If there is sampling bias in the expert counts, it is also possible that use of a limited sample size could bias the detector toward a particular definition of atmospheric river. Figure 12 shows that each EGID results in parameters that are grouped somewhat closely together in parameter space, so it is reasonable to assume that additional experts would result in new EGIDs with different groupings of parameters. There are two main reasons that we limit this study to contributions from only 8 experts: the amount of person-effort required to solicit input, and the computational expense of training the Bayesian model on each expert. In addition to the substantial person-effort invested by each additional contributor, engaging more experts would require soliciting input from experts outside of the project that funded this effort (see the *Acknowledgements* section), which would require investing in further development of the GUI (Figure 1) to port it to other systems. It seemed prudent to limit our investments in such further developments, since our initial data collection phase concluded right about the same time that the ClimateNet effort (see next paragraph) launched."

2. Another possible Bayesian model for atmospheric river detection could instead use the various algorithms presented within ARTMIP to constrain the posterior. These algorithms required the combined efforts of more scientists than were used to provide the atmospheric river counts for the current study. It would therefore be helpful for the authors to address this alternative, and to discuss the implications of using a collection of experts' opinions rather than a collection of previously generated algorithms.

This is an interesting prospect! We have added a paragraph introducing this possibility and discussing possible complications.

(orig. p16.12, cur. p21.30): "One could consider utilizing data from the ARTMIP project to constrain a Bayesian model, since each ARTMIP catalogue effectively represents each expert developer's opinion on where and when ARs can be distinguished from the background. This would greatly increase the effective number of experts, though it would likely also require a substantially more complicated Bayesian model. As noted by Ralph et al., (2019), each existing AR detection algorithm has been designed for a specific application: ranging from understanding the global hydrological cycle (Zhu et al., 1998) to understanding AR impacts in the western U.S. (Rutz et al., 2014). Forthcoming work by Zhou et al., (2020) shows that the global number of ARs detected by ARTMIP algorithms ranges from approximately 6 to 42. This is a much wider range of uncertainty in global AR count than demonstrated in this manuscript, and we hypothesize that the large upper bound is a side-effect–rather than an intended property–resulting from designing AR detectors with a focus on a particular region or impact. For example, if an AR detector designer is not particularly concerned about ARs being strictly contiguous, then global AR count would not be well constrained. If global AR count is not a reliable reflection of the AR detector designer's expert opinion, then we would need to either account for this uncertainty in the ARTMIP dataset, or we would need to formulate likelihood functions that optimize based on some other property of the ARTMIP output: ideally, properties that reflect expert opinion."

3. The large uncertainty that remains within TECA-BARD v1.0.1 even with the usage of 1,024 unique AR detectors further suggests that including more experts and/or MCMC chains per expert would help converge on a more statistically robust atmospheric river detection method. The qualitative conclusion regarding the uncertainty of the sign of correlation between ENSO phase and atmospheric river frequency is valid regardless of the remaining uncertainty within TECA-BARD v1.0.1. However, if TECA-BARD is to be used for more quantitative assessments of relationships between atmospheric rivers and climate modes (ENSO, PDO, etc.), it would be useful to constrain the uncertainty of the model even further.

We absolutely agree with the need to further reduce uncertainty, though we are not quite sure how this might be achieved. We see why the reviewer might think that including more experts or MCMC chains might reduce uncertainty. It is also possible that the opposite could be true with respect to adding more experts. We have added clarification and discussion regarding this point.

(orig. p16.6, cur. p20.34): "It therefore seems imperative to reduce uncertainty, though it is not immediately clear how that might be achieved. Adding more walkers to the MCMC calculation described in Section 2.4 would not change the underlying posterior distribution; it would only sample it more thoroughly, which would somewhat increase the spread in parameters. Adding more expert contributors (and possibly more contributions from each contributor) could have one of two main outcomes: (1) if a consensus were to emerge about AR counts, then it is possible that the EGID posterior distributions $p_j$ would start to form a 'consensus' in the combined posterior distribution, with reduced spread in the parameter space; or (2) it is possible that each new expert contribution results in a new mode appearing in the parameter space, such that uncertainty is actually increased by adding more expert contributions. Moreover, it is not clear whether the reduced parameter spread associated with outcome (1) would be desirable, since it would weight the parameter selection toward the 'consensus' of EGIDs, at the expense of suppressing 'outlier' EGIDs. The answer to this question is somewhat philosophical in nature, and the answer is likely to be application-dependent. Ultimately, physical theories about ARs may be the only reasonable way to constrain AR detection methods and therefore reduce uncertainty associated with subjective opinion."

**Specific Comments**

**Reviewer 1 Specific Comments**

Page 2, Line 29-30: I believe the ClimateNet project is working to achieve this type of dataset (spatial footprints of ARs). It is worth mentioning here and/or in the discussion how that effort might contribute to this topic and related future work.

We agree–and we certainly would have done so if the ClimateNet paper had been submitted at the time that this manuscript was submitted. We have added a brief mention of ClimateNet to the introduction, and we have added a paragraph discussing this in Section 5 (see our response to Reviewer 1's first major comment).

Page 3, Line 26: Does the user see IVT only, or are there multiple fields available to diagnose ARs?

There are multiple fields; we inadvertently omitted this information from both the text and the figure caption. The text and Figure 1 caption have both been updated with this information:

(orig. p3.26, cur. p4.4): "The meteorological plot overlays information about IVT, integrated water vapor, and the magnitude of gradients in 850 hPa equivalent potential temperature (indicative of fronts); the sample image in Figure 1 shows a screenshot of this information as it is presented to the expert contributors."

**(orig. p4.Fig1, cur. p4.Fig1):** "The expert is presented with an overlay of information about IVT (purple-yellow shading), integrated water vapor (red contours), and the magnitude of the 850 hPa equivalent potential temperature gradient (blue shading)."

Page 7, Line 2: What does $\sigma$ represent?

We have added text indicating the equation in which $\sigma$ first appears, and we have added some discussion of its meaning in the model:

**(orig. p7.2, cur. p7.17):** " We use a half-Cauchy prior for $\sigma$ (Equation 2)...$\sigma$ is the parameter controlling the width of the likelihood function, which effectively controls how far the detected counts $N_i'$ can deviate from the expert counts $N_i$ before the likelihood function indicates that a given choice of $(P, A, \Delta y)$ is unlikely compatible with the expert data; we treat it as a nuisance parameter in our model."

Section 2.3.1: Is there some way to visualize the content in this section (like Fig 3 for the previous section)? It can be challenging to follow equations only.

We have added a diagram (Figure 4 in the revised manuscript, Figure 2 in this document) that attempts to illustrate the basic concept behind the geometric constraint. In revising 2.3.1, we realized also that we had neglected to add the min suffix to $A_{\min}$ in several places in that section; we amended this. We also added reference to this diagram in the text:

**(orig. p7.19, cur. p8.14):** "Figure 4a depicts the geometric relationship between $A_{\min}$ and $P$: as $P$ increases, the maximum permissible value of $A_{\min}$ decreases."

**(orig. p8.4, cur. p9.16):** "We modify the uniform prior to be equal to 0 outside the surface defined in Equation 5 (to the right of the $A_{\min}(P, \Delta y)$ lines shown in Figure 4b)"

Section 2.3.1: Why does the f equation on page 7 have sin(delta y) but equation 5 on page 8 has sin(2*delta y)?

That was a typo in the 2nd equation; the 2 is now omitted. We note here that this typo originated from transcribing the code that defines our prior distribution in our MCMC sampling routine, which also had the erroneous value of 2. We fixed this and describe the resulting changes to the manuscript in the first section of this response. Thank you for the careful scrutiny!

Page 8, Line 1: How does grid cell size (i.e., resolution) impact these constraints?

Good question; we have added some text to clarify this.

**(orig. p8.1, cur. p9.7):** "We tighten the constraint to assert that these conditions should lead to ARs that typically have more than 1 grid cell per AR The assertion that ARs should typically consist of more than 1 grid cell is only valid if $\overline{A}$ is substantially less than the area of a typical AR. We are using MERRA2 reanalysis, with $\overline{A} = 2.5 \cdot 10^9 m^2$, which is almost two orders of magnitude smaller than the lower bound on the minimum AR size of $1 \cdot 10^{11} m^2$ (Table 1), so even the smallest possible ARs detected will consist of $\mathcal{O}(100)$ grid cells. This assertion might need to be revisited if the Bayesian model is trained on much lower resolution data. This leads to a formulation of the prior constraint that depends on the value of the latitude filter, such that only parameter combinations that satisfy the following inequality are permitted..."

Page 8, Line 23: How was statistical equilibrium determined? Autocorrelation time?

Statistical equilibrium was determined through manual inspection of the parameter traces; we also verified that we are sampling properly from the posterior by comparing the sampled posterior PDF against a brute-force calculation of the PDF. We have revised the text to make this point more clear.

[Figure]

[Figure]

Figure 2: Illustration of the geometric constraints applied to the prior distribution of parameters $P$, $A_{\min}$, and $\Delta y$. (a) A diagram depicting the interaction between percentile threshold $P$ and minimum area $A_{\min}$. Red text depicts hypothetical IVT' percentile values for individual gridboxes (gray boxes); boxes with $P$ above 0.8 are shaded in red. (b) Visualization of Equation 5 for select values of $\Delta y$, and annotation indicating regions of the $A_{\min} - P - \Delta y$ parameter space that are *a priori* implausible because they would yield no AR detections.

(orig. p8.23, cur. p10.9): "We use an informal process to assess equilibration of the MCMC sampling chains: we manually examine traces (the evolution of parameters within individual walker chains). The traces reach a dynamic steady-state after $\mathcal{O}(100)$ steps, so we expect that the chains should all be well-equilibrated by 1,000 steps. We ran a brute-force calculation of the posterior distribution on a regularly-spaced grid of parameter values (not shown) to verify that the MCMC algorithm is indeed sampling correctly from the posterior distribution, which further evinces that the MCMC process has reached equilibrium by the 1,000$^{\text{th}}$ step."

Page 9, Line 6ff: Some clarification is needed on the difference between Expert ID and Expert Group ID.

Agreed. We have added some text to clarify the relationship and distinction between them:

(orig. p9.6, cur. p11.12): "Hereafter, we use two similar and related, but distinct terms:

- **EID** - Expert ID: the identification number of a given contributor to the expert count database. EIDs are assigned in order of the mean number of ARs that the expert typically counts in a given timestep.

- **EGID** - Expert Group ID: the identification number of groups of posterior parameters obtained by training the Bayesian model on expert counts contributed by the corresponding EID (see Equation 6).

"

Page 9, Line 7: How were 128 samples acquired for each Expert ID, when some experts did not contribute that many? Unless I am misinterpreting the numbers reported in Figure 2a and line 7 on page 4 ("between 64 and 906 time slices").

We should have specified that these are MCMC samples, not count samples. We have rectified the text:

(orig. p9.7, cur. p11.7): "TECA BARD v1.0.1 uses each of the 128 MCMC samples generated for each Expert ID; with 8 Expert IDs, this gives a total of 1,024 sets of parameters used in TECA BARD v1.0.1."

Figure 7: Where is Expert Group ID shown in this figure? Caption could use some clarification.

We have added some clarifying text to the caption for the corresponding figure (Figure 9 in the revised document):

(orig. p12.Fig7, cur. p16.Fig9): "Posterior mean AR counts for each season, grouped by EGID vs median number of ARs counted by the corresponding Expert ID. There are four points (corresponding to the four seasons) for each EGID. Since Expert ID is assigned in order of increasing AR counts, the lowest EIDs occur on the left side of the graph and the highest occur on the right. Whiskers indicate the 5-95 percentile range. The dashed line shows the 1:1 line."

Page 13, Line 5: Please explain what the ELI is; what does it physically represent?

Good suggestion. We have done so:

(orig. p13.5, cur. p17.14): "ELI represents the central longitude of areas in the tropical Pacific where sea surface temperatures are warmer than the zonal mean, which–because of the weak temperature gradient approximation–is close to the longitude of maximum tropical Pacific convection. High values are associated with El Niño conditions and low values are associated with La Niña conditions."

Figure 1: Recommend increasing the size of this image. It is difficult to see the green X's, for example.

We have done so, and we will work with the GMD editorial staff to ensure that the figure remains large in the final version

Page 6, Line 6: Lower bound of parameter P is listed in text as 0.999 but in Table 1 as 0.99.

Thank you for catching this: there was an extra digit in the text version. We have amended this to 0.99

Section 2.3.1: Some of the equations don't have numbers (e.g., equations for A and f).

This was an intentional stylistic choice that follows styles we have observed in other manuscripts with a relatively large number of equations. We have chosen to only number equations that are referred to in areas of the text separate from where the equation is first introduced (e.g., 'Visualization of Equation 5 for select values of $\Delta y$' in the Figure 4 caption). We made this stylistic choice under the assumption that the GMD editorial staff would work with us to ensure that the final style of the paper conforms to GMD style guidelines.

Page 7, Line 24: I think subscript of $N_t$ should be capital ($N_T$), unless this is a different quantity.

Yes: thank you! We have made the change

Figure 5: Recommend increasing the size of this image to make text more readable.

We have increased the width of the figure from 25% of the text width to 33% of the text width

Figure 10: Expert Group colored lines could be bolder/brighter (hard to see on panels, especially yellow colors).

This is tricky, since we are visualizing 4 dimensions of data in each panel (MCMC samples from 2 parameters, $\rho_{N,\mathrm{ELI}}$, and EGID posterior contours). We have increased the linewidth of the EGID posterior contours. Hopefully this makes the contours more visible without obscuring the other dimensions of information.

**Reviewer 2 Minor Remarks**

1. It would be useful to cite and discuss recent efforts to collect domain-expert defined outlines of atmospheric rivers, as discussed in Prabhat et al. (2020); https://www.geosci-model-dev-discuss.net/gmd-2020-72/

We have done so (see the response to Reviewer 1's similar, specific remark above)

2. Typo on line 20, page 13: Change "Figures 9" to "Figure 9"

Thank you for catching this. We actually meant to refer to Figures 9a–c; we have amended the text accordingly.

3. Could the parameter space of Figure 10 be filled by using a greater number of samples from each expert? It may be illustrative to fill more of this parameter space, although I understand issues with computational limitations.

Yes, it would do so to a limited degree. We have added a brief discussion of this to the text in relation to your broader comment (number 3) about reducing uncertainty

4. The posterior PDFs for the combined EGID model shown in Figure 4 exhibit multi- modality. Do you think this multi-modality is caused by the limited number of experts used to constrain the posterior, or is there some other cause? Either way, it would be useful to discuss this within the manuscript.

Reviewer 1 had a similar question in their major comment 3; please see our response there.

5. I recommend increasing the size of Figure 5 so it is easier to read in the final manuscript. Also, it could be useful to reference the various phrases used in the flowchart within the manuscript itself.

We have increased the size of the figure, and we have added a paragraph describing the purpose of each component in the figure:

[revised manuscript text omitted]

* * *
[50]removed: all
[51]removed: all
[52]removed: 6
[53]removed: EGID 7
[54]removed: 6
[55]removed: predominantly
[56]removed: a,
[57]removed: only in Figure 11b does
[58]removed: parameter
[59]https://bitbucket.org/lbl-cascade/fastkde at commit f2564d6

colored by EGID, such that 95% of the posterior distribution for each EGID falls within the given contour; the colored contours in Figure 12 effectively outline the parameter samples for each EGID.

Parameter clusters with both positive $\rho_{N,\text{ELI}}$ and high $\Delta y$ tend to form distinct zones of points in Figure 12: clusters with relatively low [..[60] ]$P$ and relatively high $\Delta y$. Parameters with negative $\rho_{N,\text{ELI}}$ predominantly fall along [..[61] ]two lines in the $P$-$A_{\min}$ plane in Figure 12b, with the positive $\rho_{N,\text{ELI}}$ values forming [..[62] ]on the line with lower $P$ values. These separate clusters are associated with the more permissive EGIDs.

We argue that the differences in correlation coefficient between the restrictive and permissive EGIDs likely results from differences in the degree to which tropical moisture anomalies are filtered among the EGIDs. Patricola et al. (2019) show that strong El Niño events are associated with positive IVT anomalies in much of the tropics and a separate band of positive anomalies in the midlatitudes (around $30°$ latitude; their Figure 11). The positive IVT anomalies in the tropics would have no effect on the subset of AR detector parameters with high values of $\Delta y$, since these values would be aggressively filtered. This subset of parameters with high $\Delta y$–which is associated with the permissive EGIDs and positive values of $\rho_{N,\text{ELI}}$ (Figures 5b and 11b)–would then only be affected by the higher-than-average IVT in the midlatitudes. This would result in larger numbers of ARs during El Niño events. For AR detectors parameters with low values of $\Delta y$, the zone of positive anomaly in the tropics would not be totally filtered out, which increases the chances for zones of high IVT in the midlatitudes to be connected to zones of high IVT in the tropics. This could potentially result in larger-than-average, and fewer, ARs during El Niño.

**5    The Importance of Uncertainty in Feature Detection**

The results in Section 4 show that equally plausible sets of AR detector parameters can yield qualitatively different conclusions about the connection between ENSO and AR count. These results also show that the data used to constrain the AR detector parameters in TECA-BARD v1.0.1 has a huge influence on the choice of parameters and ultimately the conclusions that one might draw. Figure 10 shows that almost half of the spread in $\rho_{N,\text{ELI}}$ can be explained by the spread in expert counts used to constrain the Bayesian model. This spread results from differences in subjective opinion about what does or does not constitute an [..[63] ]AR.

There are numerous aspects of AR-related research for which TECA BARD v1.0.1 could be useful: including research on AR variability, predictability, and impacts in the observational record; and changes in AR dynamics and impacts in past and future climates. We use the ENSO-count relationship simply as a demonstration that parametric uncertainty can have a large effect on data analyses. There are numerous results in the literature for which a single AR detection method was used (or in some cases a few detection methods applied over multiple studies), including: 90% of the poleward moisture flux is associated with ARs (Zhu and Newell, 1998), 15–35% of precipitation in coastal California comes from ARs (Dettinger, 2011; Rutz et al., 2014; Guan and Waliser, 2015a; Gershunov et al., 2017; Rutz et al.,
* * *
[60]removed: $A_{\min}$

[61]removed: a line

[62]removed: separate clusters of points off the line

[63]removed: atmospheric river

[revised manuscript text omitted]